# Sublinear scaling of the cellular proteome with ploidy

G. Yahya[1,2], P. Menges[1], P. S. Amponsah ®[1], D. A. Ngandiri ®[1], D. Schulz[3], A. Wallek[4], N. Kulak[4], M. Mann ®[4], P. Cramer ®[5], V. Savage[6], M. Räschle ®[1] & Z. Storchova ®[1] ✉

Ploidy changes are frequent in nature and contribute to evolution, functional specialization and tumorigenesis. Analysis of model organisms of different ploidies revealed that increased ploidy leads to an increase in cell and nuclear volume, reduced proliferation, metabolic changes, lower fitness, and increased genomic instability, but the underlying mechanisms remain poorly understood. To investigate how gene expression changes with cellular ploidy, we analyzed isogenic series of budding yeasts from 1N to 4N. We show that mRNA and protein abundance scales allometrically with ploidy, with tetraploid cells showing only threefold increase in protein abundance compared to haploids. This ploidy-dependent sublinear scaling occurs via decreased rRNA and ribosomal protein abundance and reduced translation. We demonstrate that the activity of Tor1 is reduced with increasing ploidy, which leads to diminished rRNA gene repression via a Tor1-Sch9-Tup1 signaling pathway. mTORC1 and S6K activity are also reduced in human tetraploid cells and the concomitant increase of the Tup1 homolog Tle1 downregulates the rDNA transcription. Our results suggest that the mTORC1-Sch9/S6K-Tup1/TLE1 pathway ensures proteome remodeling in response to increased ploidy.

The majority of eukaryotic organisms are diploid (2N, with two sets of chromosomes) or haploid (1N), but polyploid cells (>2N) are common in nature. Polyploidy is found throughout the eukaryotic kingdom and plays an important role in speciation, particularly in plants[1]. Here, polyploidy increases their adaptive potential, bringing short-term success but also disadvantages, reflected by the fact that the number of established whole-genome duplications (WGDs) is low[2,3]. Polyploidy plays an important role in differentiation of multicellular organisms, where it arises in specialized organs and tissues from developmentally tightly controlled processes, or in response to stress conditions[4]. Polyploidy can also occur from an error. Unscheduled polyploidy occurs frequently in human cancers, and an estimated 37% of all cancers underwent a WGD event at some point during their progression[5]. The incidence of WGD is even higher in metastasis[6]. In addition,

increased ploidy generally impairs genome stability[7–11], promotes tumorigenesis in cancer model systems[12], and increases cell evolvability and adaptability[13]. Therefore, whole-genome doubling is an important driving force in evolution and tumorigenesis[14,15]. Despite the frequent occurrence of polyploidy and its importance in evolution and pathology, the effects of WGD on cellular physiology are only partially understood.

Budding yeasts *Saccharomyces cerevisiae* of different ploidy can be readily constructed and serve as an excellent model to study the consequences of WGD. An apparent result of increased ploidy is the linear increase in cell and nuclear volume[7,16]. Doubling the cell volume is expected to be accompanied by a 1.58-fold scaling of two-dimensional structures such as membranes, and a 1.26-fold increase in linear structures. Polyploid budding yeasts grow at slightly

[1]Department of Molecular Genetics, TU Kaiserslautern, Paul-Ehrlich Str. 24, 67663 Kaiserslautern, Germany. [2]Department of Microbiology and Immunology, School of Pharmacy, Zagazig University, Zagazig, Egypt. [3]Institute of Molecular Biology, University of Zurich, Zurich, Switzerland. [4]Max Planck Institute of Biochemistry, 82152 Martinsried, Germany. [5]Max Planck Institute for Multidisciplinary Sciences, Göttingen, Germany. [6]Department of Biomathematics, University of California at Los Angeles, Los Angeles, CA 90095, USA. ✉e-mail: storchova@bio.uni-kl.de

reduced rates compared with diploids[16–18], reaching a larger volume in a similar time span as haploid or diploid strains. Increased ploidy in yeast leads to aberrant cell cycle regulation and altered response to nutrition[16,19], lower fitness[20,21], impaired genome stability[7,22] and prominent evolvability[13]. The inherent genome instability of tetraploid yeasts can instigate ploidy reduction to diploidy during prolonged in vitro evolution, likely due to the loss of individual chromosomes[13,17]. Increased ploidy leads to increased nuclear and cell size, altered metabolism and increased genomic instability in most species analyzed so far[4,14,15]. Yet, only little is known about the underlying molecular mechanisms.

Using budding yeasts of different ploidy for transcriptome analysis showed that the expression of only a handful of genes changes disproportionately in response to ploidy, suggesting that polyploidization largely tends to maintain the balance of gene products[7,16,23]. The few genes with altered mRNA abundance encode membrane and cell wall proteins, likely reflecting an adaptation to the lower surface-to-volume ratio in larger polyploid cells[16,23]. The minimal changes observed in transcriptome raise the question of whether ploidy-dependent regulation (PDR) occurs post-transcriptionally, but to our knowledge no systematic analysis of proteome changes in cells of different ploidy has been performed so far.

A striking aspect of increased ploidy is the universal association with increased cell size, but the mechanism remains enigmatic. In the simplest reasoning, one could envision that doubling the genome will double the amount of transcripts, and therefore also the amount of produced proteins, thus leading to larger size. Indeed, cell size association with gene expression during cell cycle progression in haploid and diploid organisms revealed that gene expression scales with cell size[24,25].

Here, we set out to address the question of how yeast proteome changes with increasing ploidy. We show that while the cell volume increases nearly linearly, the global protein biosynthesis scales sublinearly with the volume and ploidy. We further demonstrate that the abundance of some proteins is differentially regulated with ploidy, and identify downregulation of the mTOR signaling to be responsible for the reduced protein biosynthesis in polyploid cells. We propose that failure to scale protein content with the cellular size might be responsible for some of the phenotypes observed in cells after whole-genome doubling.

## Results

### Proteome content scales with budding yeast ploidy

We generated a series of isogenic haploid to tetraploid strains derived from the BY4741 strain background, all with the mating type MAT**a** to eliminate the effects of the pheromone pathway (Supplementary Fig. 1a and Supplementary Table 1). Ploidy was confirmed by flow cytometry (Supplementary Fig. 1b). As expected, the volume of budding yeast cells increased linearly with ploidy[7,16], with median volumes of 48.0 fl for 1N, 82.9 fl for 2N, 146.6 fl for 3N and 181.7 fl for 4N (Supplementary Fig. 1c, d). Proliferation was slightly reduced in triploids and tetraploids, as shown previously[21] (Supplementary Fig. 1e, f).

To assess the global proteome changes in strains of different ploidy, we performed stable isotope labeling of amino acids in cell culture (SILAC) including an external spike-in standard consisting of a heavy lysine (Lys8)-labeled SILAC yeast standard, followed by liquid chromatography−tandem mass spectrometry[26]. The heavy-labeled standard represented an equal protein mix from cells of each ploidy state (1N, 2N, 3N and 4N) and was added to equal number of cells of each ploidy (Fig. 1a). The measurements provided quantitative

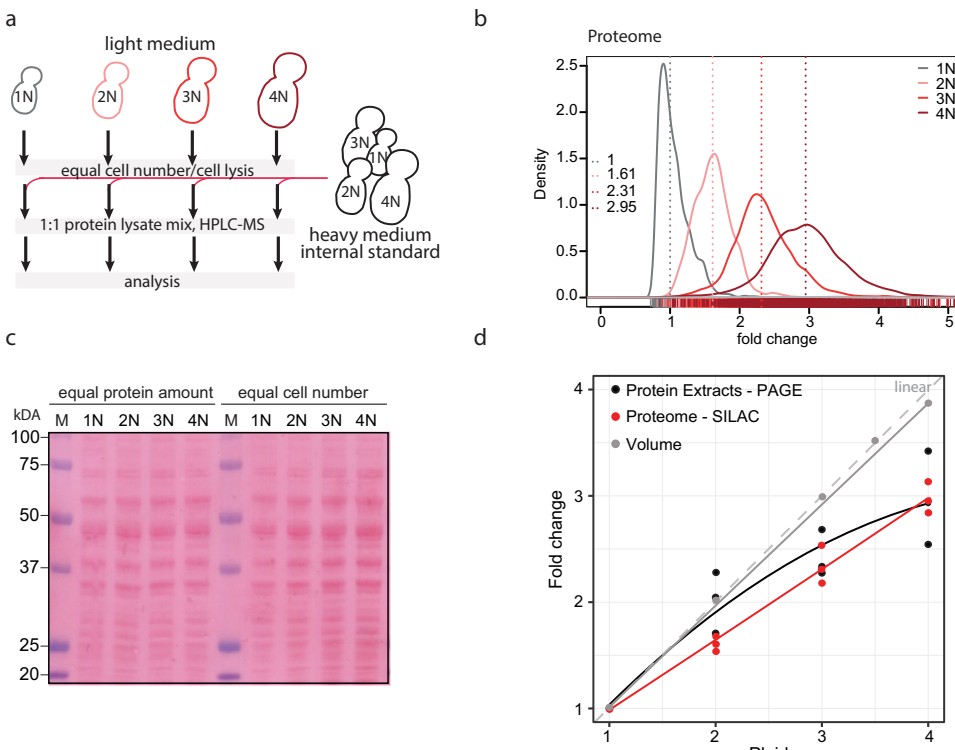

**Fig. 1 | Proteome changes in response to increasing ploidy. a** Schematic depiction of the strategy used for the proteome analysis. **b** Proteome scaling with increasing ploidy. 3109 protein groups were quantified in all ploidies. The values were normalized to the internal standard, with haploids shifted to a median at 1. All other ploidies were then shifted by the same factor. **c** Representative Ponceau staining of the total protein amount in cell of different ploidy when equal protein amount was loaded (left) and when proteins isolated from equal cell numbers were loaded (right). **d** Quantification of the volume and protein content (three independent measurements) changes with increasing ploidy with best-fit trend lines. Theoretical linear scaling is depicted in gray dashed line. Source data are provided in Supplementary Table 1 and in a Source Data file.

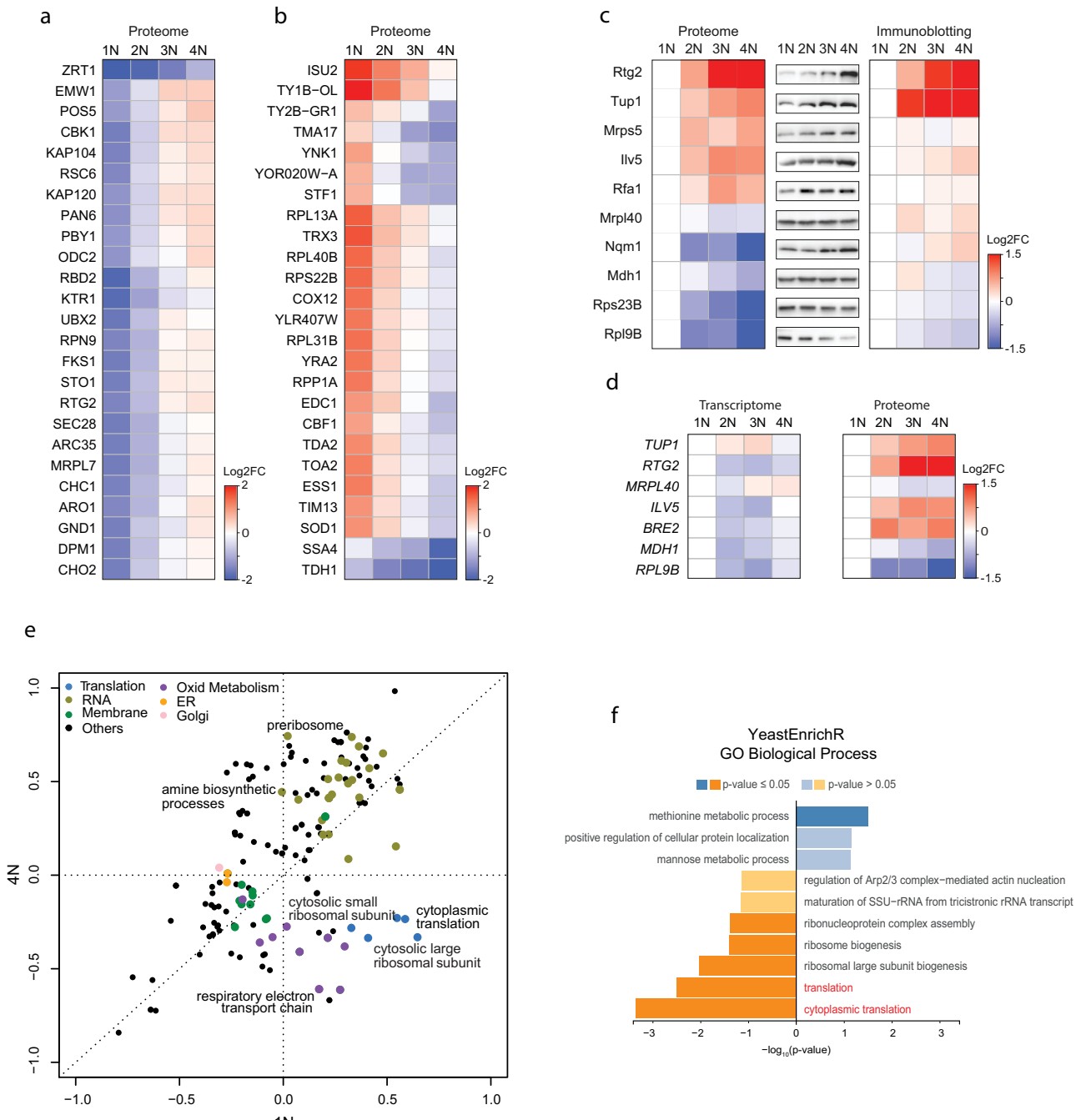

**Fig. 2 | Gene expression changes in response to ploidy. a** Top 25 proteins significantly induced with increasing ploidy. **b** Top 25 proteins significantly reduced with increasing ploidy. Depicted (in **a**, **b**) is the heat map of fold changes to the internal standards. **c** Validation of the abundance changes of selected proteins. Left: relative proteome changes with ploidy normalized to 1N; middle: representative immunoblot of the selected candidates; right: quantification of the protein abundance fold changes (FC) as determined by three biological replicates of immunoblotting, normalized to 1N. **d** Comparison of mRNA levels (by transcriptome analysis) and proteome levels (by SILAC-MS) of seven factors differentially regulated by ploidy. **e** Two-dimensional pathway enrichment analysis comparing the proteome data of 1N and 4N cells (normalized to the internal

standard). Each dot represents a pathway as determined by GO or KEGG databases. Only pathways significantly deregulated with respect to internal standard are depicted, Benjamini-Hochberg FDR < 0.02. Related pathways are color-coded. **f** Gene set enrichment analysis of the proteome regulated by ploidy. Proteins were ranked according to their combined $q$ value derived from pairwise comparisons of higher ploidy states compared to haploid (see Supplementary Data 1). YeastEnrichR was used to calculate enrichment from ranked gene lists. Enrichment of Gene Ontology (GO) terms from the category "Biological Pathway" are shown (see Supplementary Data 3 for additional categories). Source data are provided in Supplementary Tables and as a Source Data file.

information for ~70% of all verified open reading frames in each sample with high correlation between the three independent replicates (Supplementary Fig. 2a). This was also confirmed by principal component analysis (Supplementary Fig. 2b). For subsequent analysis, the protein groups were filtered to contain at least 2 valid values in at least one of the four triplicates representing each ploidy state. In total, quantitative abundance measurements were obtained for 3009 proteins in all ploidies relative to the heavy-labeled SILAC standard. The

quantification revealed that the amount of proteins per cell increased with ploidy, but it did not scale linearly (Fig. 1b, median 2N: 1.61, 3N: 2.31, 4N: 2.95). Instead, a power trend line with an exponent $b = 0.78$ fitted the date with the $R^2$ value of 0.99, indicating a good fit of the estimated trend line to the actual measurements. This "Ploidy-Dependent protein Scaling" (PDS) was validated by independent measurements of protein concentration from cell lysates, which showed similar non-linear scaling with ploidy (Fig. 1c, d and Fig. S2c).

Next, we asked how the mRNA levels change in cells of higher ploidy. To this end, we used comparative differential transcription analysis (cDTA[27]). mRNA was extracted from *S. cerevisiae* cells of different ploidy (1N, 2N, 3N, 4N) labeled with 4-thiouracil (4tU) and mixed with mRNA of the distantly related haploid fission yeast *Schizosaccharomyces pombe* labeled with 4sU (4-thiouridin) as an internal standard (Supplementary Fig. 3a, ref. 27). This allowed us to obtain data on abundance changes for 5656 mRNAs. Comparison of mRNA abundance in cells of different ploidy showed that the amount of mRNA increases with ploidy, but similarly as observed for proteins, the increase is not linear with ploidy (Supplementary Fig. 3b, c). Taken together, our data show that gene expression does not scale linearly with increasing ploidy in budding yeast. Rather, the relative abundance is lower than expected from gene copy number. This is in a striking contrast with the linear scaling of the volume with ploidy, suggesting that the cellular proteome becomes diluted as the ploidy increases.

### Subset of genes is differentially regulated by ploidy

While most proteins followed the PDS trend, we observed that several proteins were differentially regulated in cells of different ploidy. To identify the differentially regulated proteins, we first calculated the log2 fold change (FC) relative to the SILAC standard (Supplementary Data 1, the data can be visualized via a web-based application PloiDEx, see Material and Methods). We first classified all differentially expressed proteins that showed a consistent trend across all ploidies with at least 2FC of the calculated 4N/1N ratio (Fig. 2a, b and Supplementary Data 1). In addition, we found that several proteins showed >2FC 4N/1N ratio, but the trend was not consistent across all ploidies; these proteins showed often a marked abundance change between 1N and 2N state (Supplementary Fig. 4a–c). Among the proteins with ±2FC change consistently across ploidies, we found several upregulated cell wall integrity proteins (e.g., Cbk1, Prt1, Emw1, Fks1), whereas multiple mitochondrial (Isu2, Tma17, Tim13) and ribosomal (Rpl13A, Rpl31B) proteins were downregulated. Validation by immunoblotting of 10 selected proteins matched well with the proteome results (Fig. 2c). We also observed similar trends in protein abundance changes in polyploid yeast strains of a different genetic background (S288C vs. Σ1278b, Supplementary Fig. 4d, e), or when the cells were cultured in different media (synthetic vs. complete media, Supplementary Fig. 4f, g), suggesting that the PDR is a general phenomenon independent of culturing conditions. Moreover, PDR was not due to an increased volume of polyploid cells, as shown by analysis of two haploid mutants with altered cell size: *cln3Δ* that lacks a G1 cyclin, and a respiration deficient *rho*[0] mutant (Supplementary Fig. 5a, b). While the volume of these haploid mutants was comparable to that of wild type diploids and triploids (Supplementary Fig. 5c), the protein abundance of seven selected candidates did not correspond with the changes observed in polyploids (Supplementary Fig. 5d).

Polyploids are inherently unstable and thus cells with aneuploid karyotype can accumulate in the population[7,13]. It has been previously shown that the expression of several genes and pathways is differentially regulated in response to aneuploidy[28]. Although our flow cytometry analysis of the strains did not reveal a sizable subpopulation of aneuploid cells (Supplementary Fig. 1b), we asked whether the gene expression changes observed in polyploid cells resemble the cellular response to aneuploidy. Comparison of protein abundance changes of selected proteins in polyploid cells with changes previously identified

in disomic yeast strains cells[28] showed no similarity (Supplementary Fig. 6a, b). Two-dimensional pathway enrichment analysis revealed discordant pathway activation and inhibition, suggesting that aneuploidy is not a strong contributor to the observed phenotypes in polyploid cells (Supplementary Fig. 6c, d). We conclude that ploidy increase is the major determinant of the observed protein abundance changes.

Next, we asked whether the observed PDR arises due to transcriptional changes that affect mRNA levels. In agreement with previously published results[7,16,23], the expression of vast majority of mRNAs was not differentially regulated in polyploid cells (Supplementary Data File 2, PloiDEx). Only 13 mRNAs changed significantly with ploidy (±2FC), including factors involved in plasma membrane and cell wall synthesis (Supplementary Fig. 7a, b), in agreement with previous findings[7,23]. These results were also validated by qPCR (Supplementary Fig. 7c). Importantly, qPCR analysis of factors whose protein levels were strongly differentially regulated by ploidy confirmed that there are no corresponding changes in the transcripts' abundance (Fig. 2d and Supplementary Fig. 7c, d). This is in agreement with previous analyses of mRNA expression, which also revealed only marginal changes in transcripts' abundance in response to ploidy[7,16,23]. Thus, the PDR of gene expression occurs largely post-transcriptionally.

### Protein translation is reduced in polyploid yeasts

We next asked which pathways are differentially regulated in response to ploidy changes. Two-dimensional pathway enrichment comparison of differential pathway regulation in 1N and 4N revealed striking ploidy-dependent downregulation of pathways related to cytoplasmic ribosomes, translation, and mitochondrial respiration (Fig. 2e). Gene set enrichment analysis (GSEA) of the significantly deregulated proteins also confirmed that "ribosome biogenesis" and "cytoplasmic translation" were repressed with increasing ploidy, whereas vesicle trafficking and intracellular transport were activated (Fig. 2f). No Gene Ontology Biological Processes pathways were significantly enriched on transcriptome level, further confirming that most of the ploidy-dependent regulation occurs on protein level.

Proteome analysis showed that the "respiratory electron transport chain" was reduced (Fig. 2f). Consistently, the abundance of Rtg2, which plays a central role in retrograde signaling of mitochondrial dysfunction to the nucleus, was increased in cells of higher ploidy because of the stabilization of this protein (Fig. 2c and Supplementary Fig. 8a, b). Polyploid cells also proliferated poorly on media with non-fermentable carbon source or in the presence of the oxidant diamide (Supplementary Fig. 8c). This suggests ploidy-specific deregulation of mitochondrial functions, in line with recent findings in pathogenic yeast *Candida albicans*[29].

"Cytoplasmic translation" and "ribosome biogenesis" were also strongly downregulated in cells with higher ploidy (Fig. 2e, f and Supplementary Fig. 9a–c). We focused on this aspect, because it could explain the observed allometric PDS of protein content (Fig. 1b, d). Since the mRNA abundance of the transcripts of ribosomal protein genes was not reduced with ploidy and therefore could not explain the altered abundance of ribosomal proteins, we hypothesized that the production of rRNA is downregulated with increasing ploidy. Indeed, qRT-PCR revealed reduced abundance of 25S and 5.8S rRNA in polyploid cells (Fig. 3a). The abundance changes were not due to reduced number of rDNA repeats in polyploid cells, as shown by qPCR analysis (Supplementary Fig. 9d). To determine whether the reduced rRNA abundance negatively affects protein expression, we used pulse-labeling with puromycin, an aminonucleoside that is incorporated into the nascent polypeptide chain and serves as a proxy for translational efficiency. This showed that the relative translation rate increased with ploidy, but the increase was not linear, in agreement with the proteome quantification (Fig. 3b–d and Supplementary Fig. 9e). The puromycin incorporation appeared to be constant when equal amount

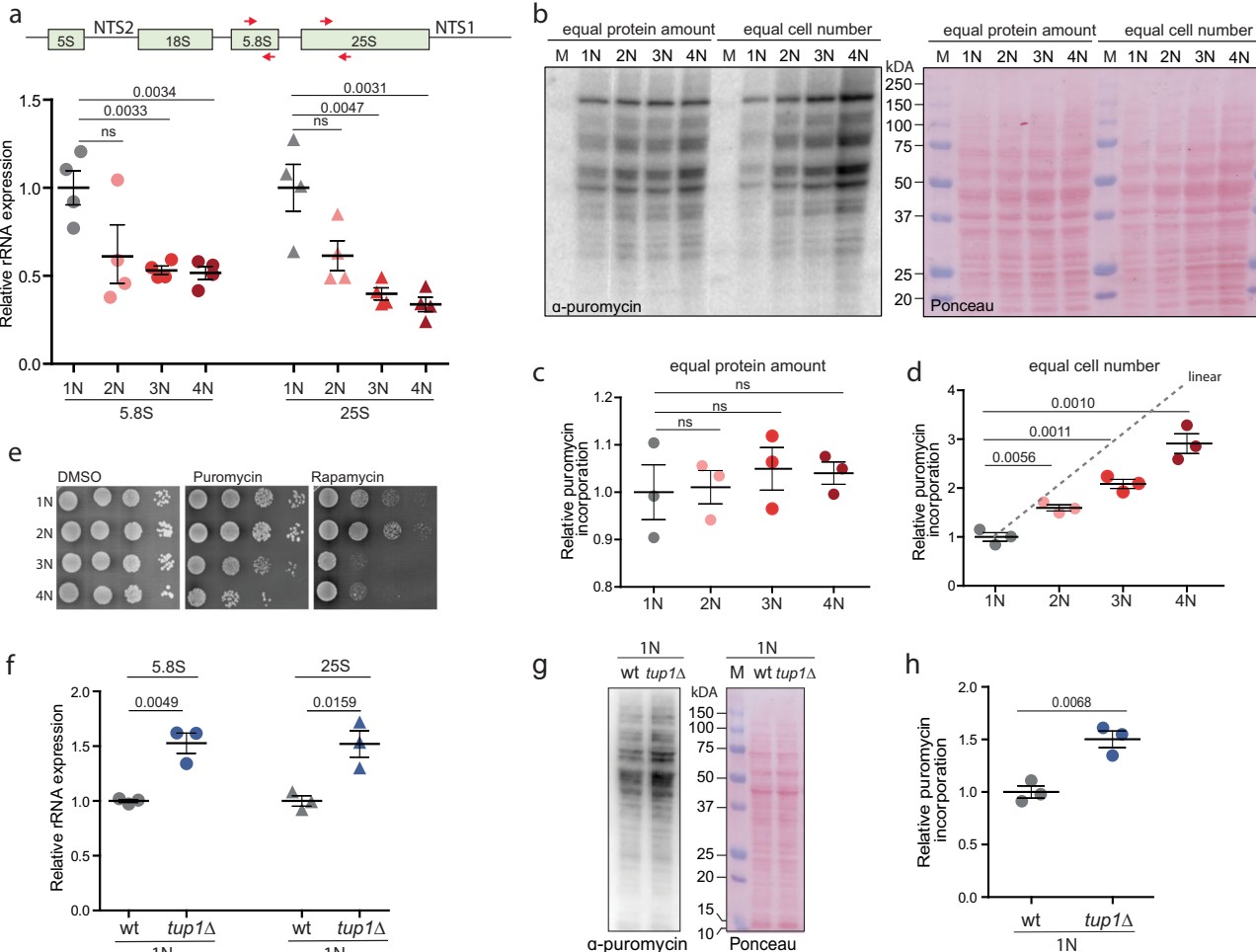

**Fig. 3 | Translation and ribosome biogenesis are downregulated in cells with increased ploidy. a** Quantification of rRNA abundance in cells of different ploidy. qRT-PCR was used in four independent experiments, the values were normalized to the expression of the housekeeping gene *ACT1* and *RPL30*. Top: schematics of the primers' localization. **b** Puromycin incorporation in cells of different ploidy. Two types of loading were applied: left: equal amount of protein lysates were loaded, right: lysates from equal cell number was loaded. Ponceau staining was used as a loading control. **c**, **d** Quantification of the relative puromycin incorporation calculated from three independent experiments; means with SEM are shown.

**e** Sensitivity of cells of different ploidy to puromycin and rapamycin treatment. **f** Quantification of rRNA in haploid wt cells and in mutants lacking *TUP1*. The experiment was performed as in **a**. **g** Representative immunoblotting of puromycin incorporation in haploid wt and *tup1Δ* cells. **h** Quantification of puromycin incorporation from three independent biological replicates. Equal amounts of protein lysates were loaded, Ponceau staining was used as a loading control. All plots show means with SEM from at least three independent experiments, the statistical evaluation was performed using the unpaired, two-tailed *t*-test, the *p* values are included. ns not significant. Source data are provided as a Source Data file.

of proteins was loaded. Additionally, the cells of higher ploidy were more sensitive to puromycin, which inhibits translation, as well as to rapamycin, an inhibitor of the master regulator of cellular metabolism mTOR, supporting the notion that the translational regulation is compromised via altered mTOR signaling (Fig. 3e).

What mechanism controls the reduced ribogenesis and translation in cells with higher ploidy? Because the rRNA levels were reduced, we hypothesized that rDNA transcription decreases with increasing ploidy. We therefore examined all differentially regulated proteins for motives commonly found in repressors (e.g., WD repeats), or for annotation indicating their involvement in rRNA transcriptional regulation. We identified Tup1, a WD40-repeats containing protein[30], whose human homolog Tle1 was previously shown to mediate rRNA repression[31]. Indeed, deletion of *TUP1* in 1N strains led to a significant increase in rRNA abundance and in puromycin incorporation rates (Fig. 3f–h), but caused no changes in the abundance of several mRNAs (Supplementary Fig. 9f). This suggests that Tup1 acts, similar to its human homolog TLE1, as a repressor of rDNA expression.

## The mTOR pathway controls the reduced translation in polyploids via Sch9-Tup1

Ribosome biogenesis in eukaryotes is regulated via the mTOR pathway. In budding yeast, Tor1 phosphorylates the kinase Sch9, a yeast homolog of human P70-S6K, to promote rRNA expression, ribosome biogenesis and protein synthesis[32]. We asked whether Tor1-Sch9 regulates the rRNA expression via Tup1. Indeed, inhibition of the mTOR pathway in haploid cells by treatment with rapamycin increased Tup1 abundance to levels observed in tetraploids (Fig. 4a, b), clearly linking the regulation of Tup1 and rRNA gene expression to the mTOR pathway. Haploid cells lacking Sch9 accumulated Tup1 independently of rapamycin treatment, suggesting that Tor1 negatively regulates Tup1 levels via the Sch9 kinase (Fig. 4c). To test whether Tup1 is a downstream target of Sch9, we constructed a haploid strain with the analog-sensitive allele of Sch9 (Sch9-as ref. 33). Inhibition with the ATP analog 1NM-PP1 increased the abundance of Tup1 and reduced its phosphorylation, as documented by altered migration in the Phos-TAG gel (Fig. 4d, e). Finally, in haploid cells, loss of *TUP1* led to constitutively high rRNA levels, while the loss of *SCH9* reduced them (Fig. 4f).

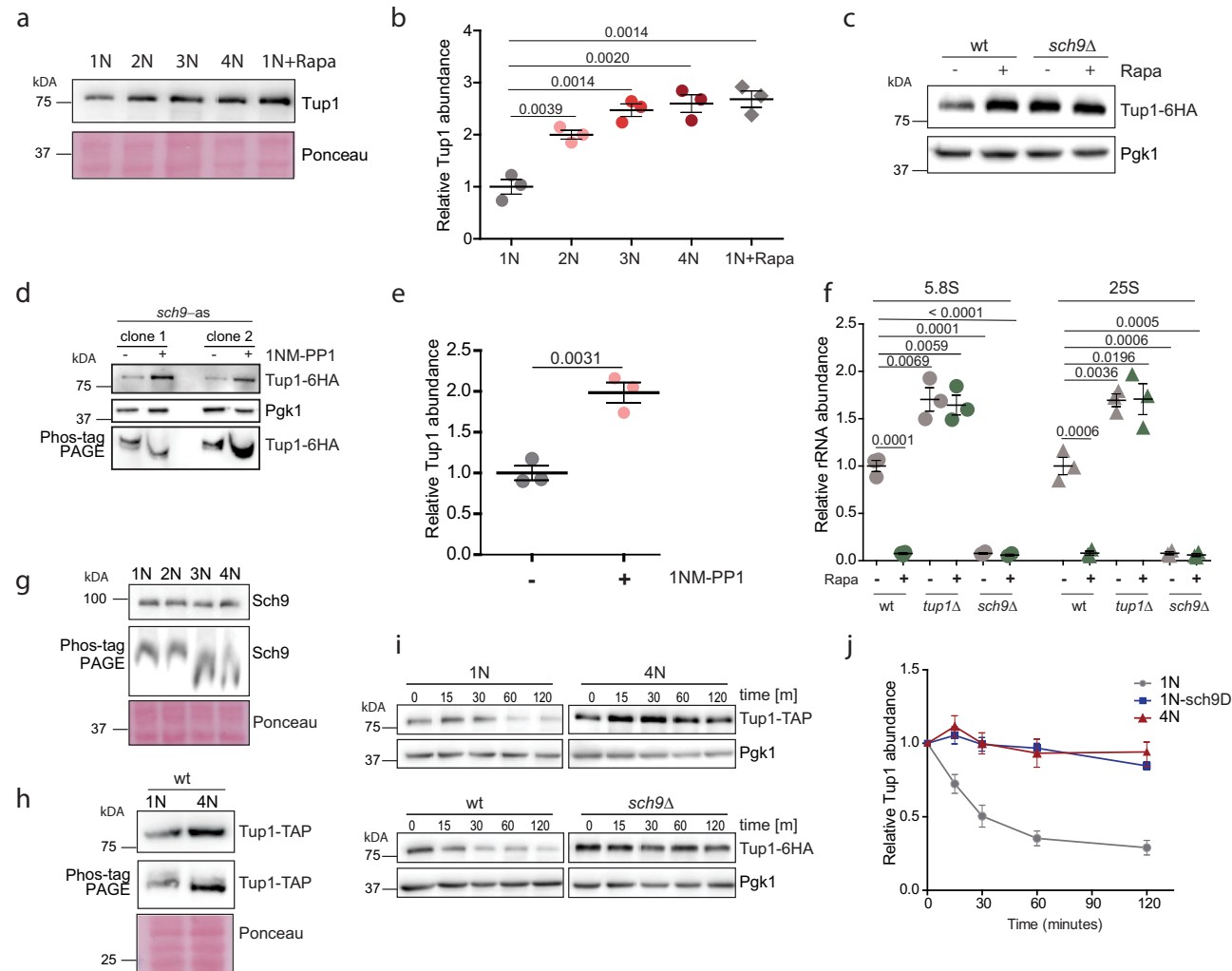

**Fig. 4 | Regulation of rRNA expression in polyploid cells via Tor1-Sch9-Tup1 signaling pathway. a** Relative abundance of Tup1 in cells of different ploidy compared to haploid cells treated with rapamycin. **b** Quantification of three independent experiments as in **a**. Mean with SEM is shown. **c** Representative immunoblot showing the abundance of Tup1 in wt and *sch9Δ* haploid cells with and without the TORC1 inhibitor rapamycin (Rapa). **d** Representative immunoblot showing the abundance of Tup1 in *sch9-as* haploid cells with and without the ATP analog 1NM-PP1, resolved on polyacrylamide gel (PAGE) with and without Phos-tag. Two different mutant clones were tested. **e** Quantification of the relative Tup1 abundance from **d**. Three independent experiments were performed, mean with

SEM is shown. **f** qRT-PCR quantification of rRNA abundance in cells lacking Sch9 and Tup1, with and without rapamycin treatment. Means and SEM of three independent experiments are shown. **g** Representative immunoblotting of Sch9 on PAGE and Phos-TAG gel in cells of different ploidy. **h** Representative immunoblotting of Tup1 on PAGE and Phos-TAG gel in cells of different ploidy. **i** Representative immunoblotting of Tup1 after cyclohexamide shut off. Different time points were collected and equal amount of lysate was loaded. **j** Quantification of three independent experiments from **h**. Means and SEM are shown. Source data are provided as a Source Data file, *p* values of the two-tailed, unpaired Student's *t* test are shown.

The observations described above suggest that the Tor1-Sch9 activity declines with increasing ploidy. Indeed, we observed decreased phosphorylation of Sch9 and Tup1 in polyploid cells, as judged by a reduced shift in migration on Phos-TAG gel. The reduced shift was similar to the shift observed upon mTOR inhibition with rapamycin, or upon treatment of the cell lysates with phosphatases (Fig. 4g, h and Supplementary Fig. 9g). Cyclohexamide shut off revealed that the stability of Tup1 increased in tetraploid cells, as well as in haploid cells lacking *SCH9* (Fig. 4i, j). We conclude that the Tor1−Sch9 activity is reduced with increasing ploidy, which leads to increased stabilization and subsequent accumulation of the negative regulator of rRNA expression Tup1. We propose that this signaling pathway contributes to the ploidy-dependent proteome scaling.

**Non-linear scaling of the proteome with ploidy in human cells**
We asked whether translation and ribosome biogenesis down-regulation also occur in human tetraploid cells. To this end, we

induced cytokinesis failure by treatment with the actin inhibitor dihydrocytochalasin D (DCD) in human diploid colorectal cancer cell line HCT116 as well as in non-cancerous cell line RPE1. This treatment induces formation of tetraploid cells[34]. Tetraploidy in human cells is largely detrimental, and usually only few cells survive[35]. By limiting dilution, we were able to isolate near-tetraploid cell lines HPT2 (HCT116 **P**ost **T**etraploid) and RPT1 (RPE1 **P**ost **T**etraploid) derived from a single parental cell after induced WGD (Supplementary Fig. 10a)[10]. Chromosome numbers and cell size increased with WGD, and the cells harbored hypo-tetraploid genome and showed increased population heterogeneity (Fig. 5a, b and Supplementary Fig. 10b, c). Strikingly, the abundance of four of six ribosomal proteins tested were reduced in HPT2 (Fig. 5c) as well as in RPT1 (Supplementary Fig. 10d). Only some ribosomal proteins were reduced in HCT116 tetraploids arising immediately after DCD treatment, suggesting that acute response to WGD may differ from chronic consequences. Of note, the RPL5 subunit, which was not reduced in any of the analyzed samples of

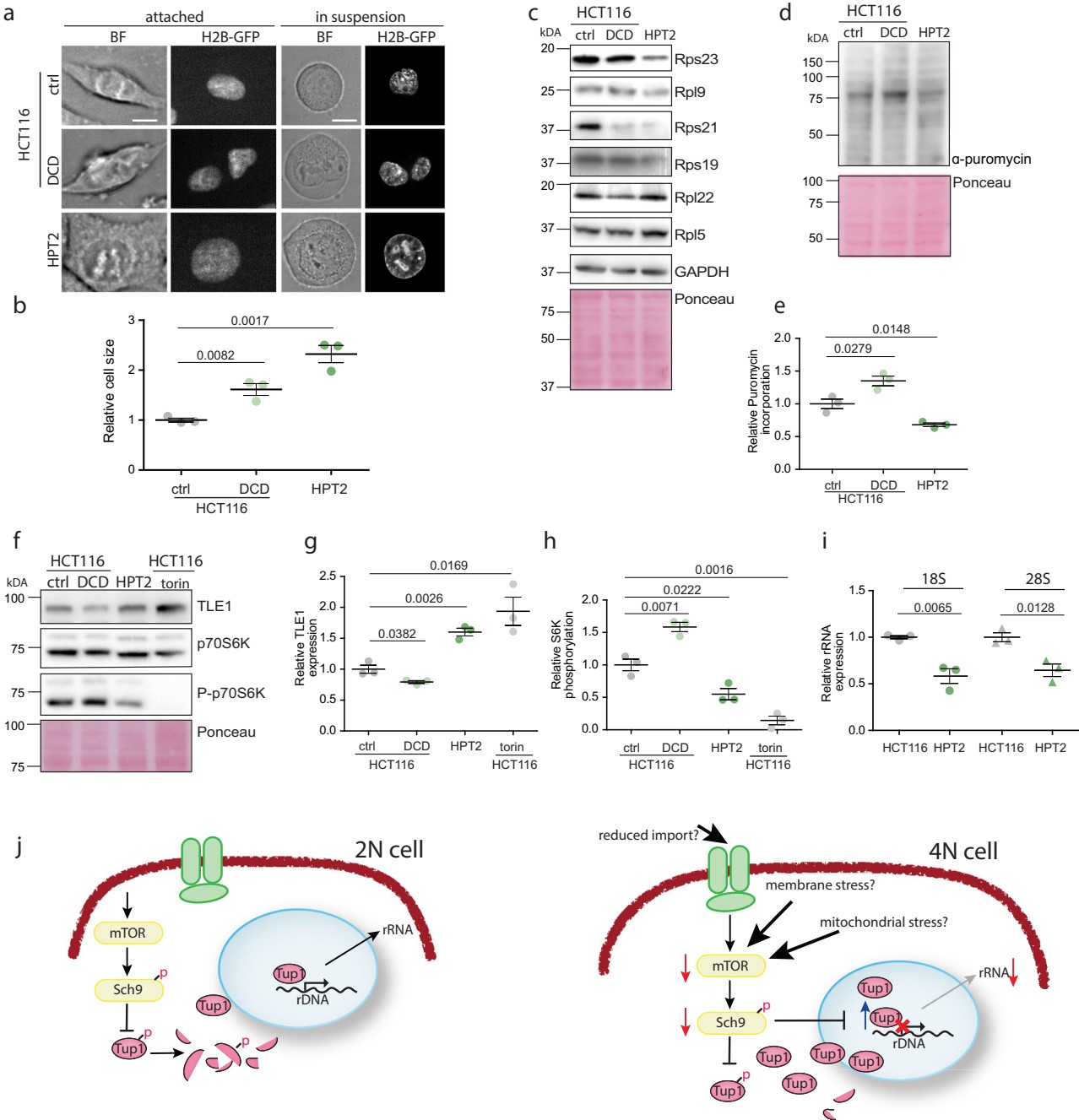

**Fig. 5 | Downregulation of ribosomal proteins and translation in human near-tetraploid cells. a** Cell size changes in response to altered ploidy. ctrl−untreated HCT116, DCD−HCT116 treated with dyhydrocytochalasin D to induce cytokinesis failure and subsequent tetraploidy; HPT2−HCT116-derived near-tetraploid cell line. The scale bar is 10 µm. **b** Relative cell size quantified from flow cytometry data. Three independent experiments were performed, means and SEM are shown. **c** Representative immunoblotting to evaluate the abundance of ribosomal proteins in response to altered ploidy. ctrl−untreated HCT116, DCD−HCT116 treated with dyhydrocytochalasin D to induce cytokinesis failure and subsequent tetraploidy; HPT2−HCT116-derived near-tetraploid cell line. **d** Puromycin incorporation in diploid, newly made tetraploid HCT116, and near-tetraploid HPT2. **e** Quantification

of the relative puromycin incorporation as shown in **d**. Three independent experiments were performed, means and SEM are shown. **f** Representative immunoblotting of Tle1 and S6K in human cells of different ploidy. **g** Quantification of Tle1; three independent experiments were analyzed, means and SEM are shown. **h** Quantification of S6K from three independent experiments, means and SEM are shown. **i** rRNA abundance in cells of different ploidy. Means and SEM of three independent experiments are shown. **j** Schematic model of gene expression regulation in polyploid cells. Source data are provided as a Source Data file. Two-tailed, unpaired Student's $t$ test was used for statistical evaluation in all relevant plots, $p$ values are shown.

higher ploidy, has a central role in ribosome biogenesis stress signaling in mammalian cells and is therefore protected from degradation. Analysis of the previously obtained transcriptome of HPT2 and RPT1[10] confirmed that these expression changes were not due to reduced transcription of ribosomal protein genes and occurred via

posttranscriptional regulation. Tetraploidy in human cells is unstable and cells with aneuploid karyotype accumulate quickly in the population, as evidenced by the variable chromosome number in HPT2 and RPT1 cell lines. Aneuploidy is known to trigger gene expression changes; therefore we asked whether aneuploid cells also downregulated

certain ribosomal proteins. To this end we used isogenic cell lines carrying an extra chromosome that we previously derived from HCT116 by microcell mediated chromosome transfer[36]. In three different cell lines with trisomy of chromosome 13 and 21, and tetrasomy of chromosome 5 the expression of ribosomal proteins was not reduced (Supplementary Fig. 10e). Thus, the changes in ribosomal protein abundance appear to be specific for cells that underwent whole-genome doubling.

Puromycin incorporation during translation was also reduced in HPT2 below the diploid levels (Fig. 5d, e). This reduction was likely mediated via the mTORC1-S6K(Sch9)-Tle1(Tup1) pathway, as Tle1 abundance was significantly elevated in HPT2 (Fig. 5f, g). Moreover, the mTORC1 activity, estimated by the phosphorylation of S6K, was reduced in HPT2, but not in the newly arising binucleated tetraploids (Fig. 5f, h). Phosphorylation of the translation initiation factor 2, α subunit (eIF2 α) was not changed, suggesting that the translational initiation was not affected by ploidy (Supplementary Fig. 10f, g). Finally, the expression of rRNA was reduced in the posttetraploid cells compared with the parental control (Fig. 5i), in agreement with the reduced mTORC1 activity. Thus, reduced mTORC1 activity and, in turn, reduced translation characterizes a cellular response to increased ploidy and may be one of the major factors responsible for the observed allometric scaling of protein levels with increased ploidy.

## Discussion

The presented proteome analysis in cells of different ploidy revealed surprising ploidy-dependent proteome changes. First, we show that protein abundance does not increase linearly with ploidy. Rather, the observed sublinear scaling appears in accordance with allometric scaling theory where metabolic rate is predicted to scale with body size with an exponent close to ¾[37,38], or, alternatively, with surface area with a scaling exponent of 2/3[39]. The PDS−ploidy-dependent scaling−may have critical implications for cell physiology. Because the genome content and cellular volume increases linearly with ploidy (Fig. S1[7,16]), allometric proteome scaling leads to cell dilution and, potentially, to a reduced ratio of DNA-binding proteins to DNA[40]. Second, we show that several individual proteins do not follow the PDS, but are rather differentially regulated in response to ploidy. We call this phenomenon policy-dependent regulation - PDR. Our findings may help explain certain phenotypes of tetraploid cells that are difficult to reconcile with the minor changes observed in the transcriptome, and provide testable hypotheses on the mechanisms underlying the ploidy-specific effects.

Strikingly, proteins related to ribosomes and translation were downregulated with increasing ploidy. The observed PDR of ribosome biosynthesis and translation also provided an explanation for the allometric proteome scaling, which we propose is regulated by the mTORC1-Sch9-Tup1 axes (Fig. 5j). Tup1 is a general transcription repressor that affects promoter accessibility[41]. We found that Tup1 becomes stabilized by reduced Tor1-Sch9 activity in polyploids, resulting in reduced rRNA abundance. How exactly Tup1 affects rRNA abundance in budding yeast, and whether the effect is direct or indirect, needs to be addressed in the future. To this end, the impacts of Tup1 overexpression on rRNA synthesis and processing, as well as the binding of PolI at the ribosomal DNA promoter in cells of different ploidy should be analyzed. The abundance of rRNA is limiting for ribosome biogenesis and thus the regulation of rRNA synthesis is central to overall ribosome synthesis. In agreement, the Tup1 homolog in humans, TLE1, previously shown to regulate rRNA expression[31], is also stabilized in tetraploid human cells, suggesting that the ploidy effect on rRNA expression may be conserved in eukaryotes. In future, global proteome changes in response to increased ploidy should be evaluated in other species as well.

Why is the rRNA expression reduced in polyploids? Ribosome biosynthesis is tightly regulated in response to environmental stress, and indeed stressors such as heat shock, osmotic shock, and nutrient deprivation reduce the expression of ribosomal proteins[42] and rRNAs[43]. The reduced translation and rRNA expression could be a direct fitness cost of increased cell volume. The decreased surface-to-volume ratio may impair nutrient uptake, which in turn reduces mTOR activity and forces downregulation of protein synthesis. Tetraploid yeast cells are sensitive to starvation[19] and often undergo ploidy reduction[13,17,22]. Finally, stable tetraploid cells evolved for 1000 generations were smaller than the original cells, although they retained the tetraploid karyotype[44]. These findings imply that cell size might be a limiting factor causing selection pressure to reduce ploidy or cell size, or both. However, our results suggest that cell volume changes alone do not lead to altered abundance of selected proteins that were affected by ploidy changes (Fig. S6). Therefore, other mechanisms may play a role. For example, membrane stress due to abnormally large volume, and mitochondrial defects are known to alter mTOR activity and therefore may contribute to the observed PDR and PDS[45]. Future research should dissect the regulatory mechanisms and upstream triggers responsible for the proteome changes in cells of different ploidy.

Another possibility is that downregulation of rDNA transcription could enhance adaptation to increased ploidy to prevent aberrant homologous recombination within the transcriptionally active rDNA repeats and R-loop formation. This hypothesis is also supported by previous finding that the kinase Sch9 is required for the evolution of genomically stable yeast tetraploids[44]. Genomic stability is impaired in cells with higher ploidy, and recombination rates increase significantly in budding yeast tetraploids[7]. Because formation of R-loops threatens genome stability and increases recombination, frequent transcription of rDNA could be fatal to already compromised genomic stability in tetraploid cells.

Finally, an important observation is that PDR occurs largely on proteome level, via changes in protein stability. Proteome changes can occur rapidly and likely respond to more subtle environmental changes, while stronger stress or prolonged in vitro evolution would be required to rewire transcriptional patterns. Interestingly, expression of the same factors and pathways identified here was altered at the transcriptional level after autopolyploidization in plants[46]. The expression of ribosomal and mitochondrial genes is also preferentially downregulated during evolution after whole-genome duplication in Salmonidae[47]. Together, our findings suggest global cellular changes triggered by increased ploidy that allow metabolic adjustments to ensure survival of cells after whole-genome doubling.

## Methods
### Yeast media and culture
YP medium containing 1% yeast extract, 2% bacto-peptone, 2% dextrose, galactose or glycerol, respectively, were used. The synthetic drop-out (SC) medium consists of 5 g/l $(NH_4)_2SO_4$, 2 g/l $KH_2PO_4$, 0.5 g/l $MgSO_4 \cdot 7H_2O$, 0.1 g/l $CaCl_2 \cdot 2H_2O$, 0.02 g/l $FeSO_4 \cdot 7H_2O$, 0.01 g/l $ZnSO_4 \cdot 7H_2O$, 0.005 g/l $CuSO_4 \cdot 5H_2O$, 0.001 g/l $MnCl_2 \cdot 4H_2O$, 1 g/l yeast extract, 10 g/l glucose, 0.5 ml/l 70% $H_2SO_4$. For drug resistance selection, 200 μg/ml G418, 100 μg/ml ClonNat, and 300 μg/ml Hygromycin, respectively, were supplemented in YPD. For drop dilution, YPD plates were supplemented with 10 ng/ml Rapamycin, 500 ng/ml Tunicamycin, 25 μM Puromycin, or 0.25 μM Diamide. For measurements of the protein half-life, the indicated strains were grown exponentially in YPD, then Cycloheximide was added to a final concentration of 50 μg/ml to inhibit protein synthesis, this point was defined as time zero. Cell aliquots were collected at the indicated time points for protein extraction and western blotting. For experiments employing the sch9-as allele, 10 nM of 1NM-PP1 was used to inhibit the kinase activity of sch9-as.

## Yeast strain construction

To obtain yeast isogenic strain of different ploidies suitable for analysis, we used the strains of S288C genetic background BY4741 and BY4742. *LYS2* gene was deleted using ClonNat cassette to enable labeling with heavy lysine for SILAC experiments. BY4741 was transformed with a plasmid carrying the HO gene under the control of galactose-inducible promoter. Diploid strain was created by mating Mat**a** and MAT**α** haploid strains. Upon induction of HO expression, single diploid colonies that become mating-competent were selected and used for further mating to create triploid and tetraploid strain (Supplementary Fig. 1a). All strains were switched by HO expression to MATa mating type and the HO-expressing vector was removed by treatment with 5-FOA.

Gene knockout strains were either retrieved from the available libraries deposit in public repository Euroscarf, or generated by homologous recombination using PCR products containing a drug cassette (kanMX6, ClonNat, hygMX) and 40 bp sequences flanking the target gene. Tagged-protein strains were either retrieved from the publicly available library (Euroscarf) or generated by integrating a cassette containing a protein tag and a drug resistance cassette at the C-terminus. PCR products were transformed into a haploid strain, or into diploid strains and the heterozygous diploids were sporulated and dissected to select for haploids with drug resistance. Subsequently, the procedure of polyploid construction was performed. For construction of strains with analog-sensitive *sch9* allele, pRS414::sch9as (T492G) was used (a gift from Robbie Loewith, University of Geneva). The list of used strains is in Supplementary Table 1.

## Human cell line culture

HCT116 (Sigma-Aldrich, ACC NO: 91091005), the postetraploid derivative HPT2 and the aneuploid cell lines, as well as RPE1 (kind gift of Steven Taylor) and postetraploid derivative RPT1, were cultured in DMEM (Life Technologies) with 10% fetal bovine serum (Sigma-Aldrich) and 1% penicillin-streptomycin-glutamine (Life Technologies). Cells were incubated at 37 °C, 5% CO$_2$ and passaged twice a week using Trypsin-EDTA (0.25%) (Life Technologies). Cells were tested for mycoplasma contamination using the MycoAlert Mycoplasma Detection Kit (Lonza), according to the manufacturer's instructions.

## Induced cytokinesis failure in human cells

Cells were treated with 0.75 μM actin depolymerizing drug dihydrocytochalasin D - DCD (Sigma) for 18 h. Subsequently, the drug was washed out 3x using prewarmed PBS. Cells were further cultured in drug-free medium for indicated time or immediately harvested for further experiments.

## Construction of post-tetraploids (WGD survivors) and aneuploids

HCT116 and RPE1 were treated with 0.75 μM of the actomyosin inhibitor DCD (Sigma) for 18 h. The cells were then washed, placed into a drug-free medium and subcloned by limiting dilution in 96-well plates (0.5 cell per well). After clone expansion, cells were harvested for flow cytometry to measure the DNA content. Subsequently, the genome was analyzed by SNP array analysis and multicolor fluorescent in situ hybridization to validate the tetraploidy. For further details see ref. 10.

To obtained trisomic cells, microcell mediated chromosome transfer was performed[36]. Briefly, A9 mouse cells containing individual human chromosomes marked with a gene for antibiotic resistance were treated for 48 h with 10 μg/ml colchicine at 37 °C to induce micronucleation, and the micronuclei were isolated by repeated centrifugation and filtration. The isolated micronuclei were fused with the recipient cell lines and selected in DMEM with the appropriate antibiotic. Single-cell colonies were collected, expanded and sequenced. Here, HCT116 with extra copy of chromosome 13 (HCT116 13/3) and 21 (HCT116 21/3), and with two extra copies of chromosome 5 (HCT116 5/4) were used.

## RNA isolation

In total, 25 OD600 of yeast cells were collected from YPD exponential cultures, cell pellets were washed twice with ddH2O, then flash frozen in liquid nitrogen. In total, 1 ml of trizol was added to frozen cell pellet and left on ice, cells were then resuspended and transferred to RNase-free screw cap Eppendorf containing 200 μl of acid washed nuclease free glass beads. Cells were disrupted by bead beating the tubes using the FastPrep-24™ 5G, 3 cycles of 20 s at speed 6 m/s and 2 min in between each cycle. In total, 200 μl chloroform was added, vortexed 15 s, then incubated at room temperature for 5 min. After the incubation the samples were spinned down at 13,000 rpm, 5 min, 4 °C. The supernatant was recovered into a fresh tube, and repeated trizol/chloroform extraction with 1 ml trizol and 400 μl chloroform was performed. Recovered supernatant in a fresh tube was precipitated by adding 0.5 ml of isopropanol and incubating on ice for 15 min. The precipitated RNA was pelleted by centrifugation at 4 °C, washed twice with 1 ml 70% EtOH, and air-dried for 10–20 min. The pellet was dissolved in an appropriate amount of RNase-free water depending on the pellet size. The quality was determined by measuring the RNA concentration and checking the quality by Nano Drop, and by agarose gel electrophoresis.

Total RNA was isolated from human cells (1 × 10$^6$ cells) using the TRIZOL reagent according to the manufacturer's instructions (Thermo Fisher Scientific). RNA quality of each sample was confirmed with by Nano Drop and agarose gel electrophoresis.

## RT-qPCR

To assess the mRNA levels, total mRNA was isolated using a Qiagen mRNeasy mini kit according to manufacturer's protocol. Next, reverse transcription using Anchored–oligo(dT) and Roche Transcriptor First Strand cDNA synthesis Kit (Cat no. 04 379 012 001) was performed to obtain cDNA. Quantitative PCR was performed using specific primers and SsoAdvanced Universal SYBR Green Supermix (Bio-Rad, USA). Melting curve analysis was performed to confirm the specificity of amplified product. Human cells' RNA was spiked with TATAA Universal RNA Spike II control (TATAA Biocenter AB, Sweden). mRNA expression of each sample was normalized to control housekeeping gene *RPL30* and the spike (human RNA) or *ACT1* (yeast RNA). The list of used primers is in Supplementary Table 3.

## Quantification of 25S ribosomal DNA by qPCR

DNA was extracted from logarithmic phase growing yeast cells using the QIAamp® DNA Mini Kit according to the manufacturer's protocol. rDNA was quantified from 10 ng of total DNA using SYBRGreen Mastermix and previously described 25S rDNA specific primers[48] (Supplementary Table 3). *ACT1* was used as a housekeeping gene for control. Thermo-cycling was performed with the following run conditions: initial denaturation at 95 °C for 3 min, followed by 40 cycles of 15 s denaturation at 95 °C and amplification at 58 °C for 30 s. A final heating gradient was performed from 58 °C to 65 °C for 5 s and then to 95 °C at 0.5 °C/cycle. Haploid yeast strain treated with rapamycin to reduce the number of the rDNA repeats was used as a control.

## Fluorescence microscopy

Exponentially growing yeast cells were imaged without fixation.

Human cells were seeded and treated when required in a glass-bottom 96-black well plate. The cells were then fixed using ice-cold methanol, permeabilized with 3% Triton X-100 in PBS and blocked in blocking solution (5% Fetal Bovine Serum + 0.5% Triton X-100 + 1% Na3N in PBS). The DNA was stained DAPI or Vybrant DyeCycle™ Green for 1 h in RT. Before imaging, cells were washed 4x with PBS.

Imaging was carried out on a spinning disc system comprising of inverted Zeiss Observer.Z1 microscope, Plan Apochromat 63x magnification oil objective, 40x magnification air objective or 20x magnification air objective, epifluorescence X-Cite 120 Series lamp and lasers: 473, 561 and 660 nm (LaserStack, Intelligent Imaging Innovations, Inc., Göttingen, Germany), spinning disc head (Yokogawa, Hersching, Germany), CoolSNAP-HQ2 and CoolSNAP-EZ CCD cameras (Photometrics, Intelligent Imaging Innovations, Inc., Göttingen, Germany).

## Flow cytometry

For DNA content, 1 $OD_{600}$ of yeast cells growing exponentially in YPD was collected and washed with $ddH_2O$. Cells were fixed in 1 ml 70 % (v/v) ethanol rotating at 4 °C overnight. Fixed cells were then pelleted at 5000 rpm for 2 min and washed with $ddH_2O$. Cells were subsequently resuspended in 500 µl FxCycle™ PI/RNase staining solution (Life Technologies, #F10797), incubated at room temperature in the dark for 30 min and then stored at 4 °C for 72 h or processed for next step. Samples were sonicated at 40% amplitude for 15 s and analyzed on an Attune™ Flow Cytometer. Data analyses was performed using the FlowJo™ software, version 10. Human cells were trypsinized and incubated in cold PBS supplemented with 5% fetal calf serum (Sigma-Aldrich; PBS-FACS). DNA was stained either by propidium iodide (PI); the cells were fixed in cold 70% ethanol added dropwise while vortexing, and incubated on ice for 30 min. Cells were centrifuged and pellets were washed twice with PBS-FACS. In total, 50 µl RNase A solution (100 µg/ml in PBS) was added to the pellet, followed by staining with 400 µl PI solution (50 µg/ml in PBS) per million cells. Cells were then incubated for 10' at 25 °C. For Hoechst staining, pellets were incubated in the dark with 10 mg/ml Hoechst 33358 for 15' at 4 °C. Data acquisition was performed using the ATTUNE NxT flow cytometer (Thermo Fisher). Data analysis was performed using the FlowJo software. Gating strategy: an SSC-A/FSC-A gate was set in order to exclude cell debris, and an FSC-A/FSC-H gate was then set in order to exclude doublets.

## Cell size measurement

Cell volume of budding yeast was determined from microscopy bright field images of exponentially growing cultures using BudJ plugin according to ref. 49. To determine the cell volume of human cells, exponentially growing cells were harvested and washed with PBS twice. Forward scatter (FSC) was measured by ATTUNE NxT Flow Cytometer (Thermo Fisher) and used as the indicator of cell size. Three biological replicates were examined and plotted.

## Cell growth for mass spectrometry

The cells were grown in SILAC synthetic drop out medium exponentially at a room temperature. The cell amount in each culture with different ploidy was counted using Burker chamber and equivalent amounts of cells of each ploidy were harvested and washed with PBS. To prepare the SuperSilac, the cells were grown in SILAC heavy medium, counted and mixed 1:1:1:1.

## Yeast sample preparation for proteomic analysis

Sample preparation was done as described as in ref. 26. Briefly, cells were lysed in SDS lysis buffer (5% SDS, 100 mM dithiothreitol, 100 mM Tris pH 7.6), boiled for 5 min at 95 °C and sonicated for 15 min (Bioruptor Sonicator, 20 kHz, 320 W, 60 s cycles). Insoluble remnants were removed by centrifugation at 16,000 × g for 5 min and clarified protein extract was transferred to a 30 kDa MW cut-off spin filter (Amicon Ultra 0.5 ml Filter, Millipore). SDS was completely replaced by repeated washing with 8 M urea. Cysteines were then alkylated using excess amounts of iodoacetamide. Proteins were then proteolytically digested overnight using LysC endoprotease (1:50 w/w enzyme to protein). Peptides were eluted and desalted using $C_{18}$ StageTips.

## Liquid chromatography coupled mass spectrometry

MS-based proteomic measurements were performed as in ref. 26. Briefly, ~2 µg of desalted peptides were loaded and analyzed by linear 4 h gradients. The LC system was equipped with an in-house made 50-cm, 75-µm inner diameter column slurry-packed into the tip with 1.9 µm $C_{18}$ beads (Dr. Maisch GmbH, Product Nr. r119.aq). Reverse phase chromatography was performed at 50 °C with an EASY-nLC 1000 ultra-high-pressure system (Thermo Fisher Scientific) coupled to the Q Exactive mass spectrometer (Thermo Fisher Scientific) via a nano-electrospray source (Thermo Fisher Scientific). Peptides were separated by a linear gradient of buffer B up to 40% in 240 min for a 4-h gradient run with a flow rate of 250 nl/min. The Q Exactive was operated in the data-dependent mode with survey scans (MS resolution: 50,000 at $m/z$ 400) followed by up to the top 10 MS2 method selecting ≥2 charges from the survey scan with an isolation window of 1.6 Th and fragmented by higher energy collisional dissociation with normalized collision energies of 25. Repeated sequencing was avoided using a dynamic exclusion list of the sequenced precursor masses for 40 s.

## MS-data analysis

Raw files were analyzed by MaxQuant software version 1.6.3.3 and searched against the *S. cerevisiae* Uniprot FASTA database (UniProt ID: UP000002311). Lysine-0 (light) and Lysine-8 (heavy) were used as SILAC labels. Cysteine carbamidomethylation was set as a fixed modification and N-terminal acetylation and methionine oxidation as variable modifications. LysC/P was set as protease and a maximum of two missed cleavages was accepted. False discovery rate (FDR) was set to 0.01 for peptides (minimum length of 7 amino acids) and proteins and was determined by searching against a generated reverse database. Peptide identification was performed with an allowed initial precursor mass deviation up to 7 ppm and an allowed fragment mass deviation of 20 ppm.

## Analysis of proteome data

MS data was analyzed in R or the software Perseus, v1.6.8. Identified protein groups were filtered to remove contaminants, reverse hits and proteins identified by site only. SILAC light/heavy ratios were calculated and transformed to log2 scale. Next, protein groups which were quantified more than two times in at least one group of replicates (N = 3) were kept for further processing, resulting in a set of 3109 protein groups in total.

For the MS quantification of the total protein amount in 1N, 2N, 3N and 4N yeast cells (Fig. 1b, d), L/H ratios from MaxQuant were filtered for two valid values in at least one ploidy. To ease comparison between different ploidies, the distribution of L/H ratios of haploid cells (1N) was shifted to be centered at 1 and all other ploidies were shifted by the same factor.

To identify proteins with ploidy-specific regulation (PDR), median centered L/H ratios of replicate groups of different ploidies were analyzed by a modified Student's t test implemented in Perseus, which provides permutation-based FDR control[50]. Pairwise two-tailed t-tests were calculated for 2N/1N, 3N/1N and 4N/1N as well as a combined score, and q values derived from permutation of the data across all ploidy states are reported in Supplementary Data 1 (FDR < 0.05, S0 = 0). This allowed identification of proteins with the highest and lowest median subtracted L/H ratio difference between 4N and 1N (log2 FC > 2). To identify proteins with a consistent trend across ploidies, the proteins were further filtered to remove any proteins with a log2 FC difference between two consecutive ploidy states exceeding 1 (see Supplementary Data 1, column *is_smooth*).

As a complementary approach to identify PDR proteins, median centered L/H intensity ratios were fitted to a linear model using the R function lm(ratio L/H~ploidy). At least four data points per protein group were required to calculate the slope (coefficient 2), two-sided

$t$-statistic and $p$ value [Pr(>|t|)], otherwise NA was reported (Supplementary Data 1).

To identify ploidy regulated pathways, 2D annotation enrichment analysis was performed in Perseus (GOBP, GOCC, permutation-based Benjamini-Hochberg FDR threshold <0.02). GSEA was performed with the webtool YeastEnrichR. As input, proteins were scored according to the statistics obtained from the linear model (see above). Enrichment scores were calculated separately for upregulated and downregulated proteins. EnrichR-combined score of downregulated pathways were multiplied by −1 and visualized together with those of the upregulated pathways in a single graph (Fig. 2f). For EnrichR-combined scores (product of the $p$ value resulting from the Fisher exact test and the $z$-score of the deviation from the expected rank) and associated $p$ values see Supplementary Data 3.

## PloiDEx−supplementary plotting application

To allow users the plotting of our combined datasets Ploidy-Dependent Expression (PloiDEx), a suave.io based application has been written in the functional programming language F# (4.5.2). The console application was built in Visual Studio 2019 for the.NET Framework 4.6.1. It allows interactive plotting of both the proteome and transcriptome data, which have been normalized as described above, in the user's default browser using the graphing library plotly.js in the form of heatmaps and profile plots. For comparability, the transcriptome has been matched to the proteome dataset. Filtering for outliers by a threshold and smoothing as described above was implemented. Additionally, the application allows the filtering of the dataset by GO cellular compartment or biological processes to plot the associated values from the database. For further information about the functionality of the application, used packages and libraries a Readme.pdf has been written, which together with the release and dependent packages is available at https://github.com/PMenges/PloiDEx.

## Dynamic transcriptome analysis (DTA)

We used the isogenic strains of different ploidies that were transformed with plasmid YEpEBI311 (2 μm, *LEU2*) carrying the human equilibrate nucleoside transporter hENT1. cDTA was performed as previously described[51]. Briefly, *S. cerevisiae* cells were grown in SD medium overnight, diluted to an $OD_{600}$ of 0.1 the next day and grown up to a mid-log phase ($OD_{600}$ of 0.8) and labeled with 4-tU (Sigma, 2 M in DMSO) for 6 min at a final concentration of 5 mM. *Schizosaccharomyces pombe* cells were grown in YES medium (5 g/liter yeast extract; 30 g/liter glucose; supplements: 225 mg/liter adenine, histidine, leucine, uracil, and lysine hydrochloride) and labeled with 4sU (50 mM in $ddH_2O$) for 6 min at a final concentration of 0.5 mM. A final concentration of 5 mM of 4tU was used. Cells were harvested via centrifugation and cell pellets re-suspended in RNAlater solution (Ambion/Applied Biosystems). The cell concentration was determined using a Cellometer N10 (Nexus) before flash freezing the cells in liquid nitrogen. Total RNA was extracted with the RiboPure-Yeast Kit (Ambion/Applied Biosystems), following the manufacturer's protocol. Labeled RNA was chemically biotinylated and purified using strepatavidin-coated magnetic beads. Labeling of samples for array analysis was performed using the GeneChip 3′IVT labeling assay (Affymetrix) with 100 ng input RNA. Samples were hybridized to GeneChip Yeast Genome 2.0 microarrays following the instructions from the supplier (Affymetrix).

## Dynamic transcriptome analysis (cDTA)

Analysis of cDTA data was carried out as described[51] using R/Bioconductor and the DTA package. Briefly, array probes that cross-hybridized between *S. pombe* and *S. cerevisiae* were removed from the analysis. The labeling bias was performed as described[27] and proportional rescaling of the *S. cerevisiae* microarray intensities to the internal

*S. pombe* standard was performed to obtain global expression FC of mRNA[51]. Differential gene expression analysis was performed using the R/Bioconductor package "Limma" and multiple testing correction was done using FDR.

## Protein isolation from budding yeast and human cells

Exponentially growing yeast cells were harvested, resuspended in 100 μl lysis buffer and incubated 10 min on ice. In total, 40 μl 100% TCA was added and incubated for 10 min on ice. Precipitated proteins were spun down for 10 min at 4 °C, 13,000 rpm. The pellet was washed with 1 ml ice-cold acetone, dried at 50 °C for 5–20 min and resuspended in 50 μl 2x Laemmli buffer. Protein lysates were boiled for 5 min at 96 °C. Pelleted human cells were lysed in RIPA buffer with protease inhibitor cocktail (Pefabloc SC, Roth, Karlsruhe, Germany), then sonicated by ultrasound in a water bath for 15 min. Cell lysate was spun down at 13,600 rpm for 10 min at 4 °C on a table-top microcentrifuge (Eppendorf, Hamburg, Germany). In total, 1 μl was used to determine protein concentration using Bradford dye at 595 nm wavelength. Subsequently, the lysates were mixed with 4x Lämmli buffer with 2.5% ß-mercaptoethanol and boiled at 95 °C for 5 min.

## Immunoblotting

Cell lysates were separated by SDS-PAGE using 10% or 12.5% gels. Protein size was estimated using the PrecisionPlus All Blue protein marker (Bio-Rad, USA). Gels were incubated in Bjerrum Schafer-Nielsen transfer buffer and proteins were transferred to a water-activated nitrocellulose membranes (Amersham Protran Premium 0.45 NC, GE Healthcare Life Sciences, Sunnyvale, USA) using semi-dry transfer (Trans-Blot® Turbo™, Bio- Rad, USA). Membranes were stained in Ponceau solution for 5 min and scanned to be used as a loading control. Next, membranes were blocked in 5–10% skim milk in TBS-T (Fluka, Taufkirchen, Germany) for 1 h in RT. After blocking, membranes were incubated in respective primary antibodies diluted in 1% Bovine Serum Albumin or 5% skim milk overnight at 4 °C with gentle agitation. Further, the membranes were rinsed 3 × 5 min with TBS-T, incubated 1 h in RT with HRP-conjugated secondary antibodies (R&D Systems), and followed by rinsing 3 × 5 min with TBS-T. Chemiluminescence was detected using ECLplus kit (GE Healthcare, Amersham™) and Azure c500 system (Azure Biosystems, Dublin, USA). The linear range was determined using the digital imaging software of the Azure c500, AzureSpot Pro, which notifies when the acquired image contains saturated bands. Protein band quantification was carried out with ImageJ (National Institutes of Health, http://rsb.info.nih.gov/ij/). A list of used antibodies can be found in Supplementary Table 2.

## Phos-tag™ SDS-PAGE

Phospho-affinity gel electrophoresis for mobility shift detection of phosphorylated proteins was performed using Phos-tag acrylamide 4.5% (w/v) running gels polymerized with 25 μM Phos-tag acrylamide (FUJIFILM Wako Pure Chemical Corporation) and 50 μM MnCl2. Gel running and transfer conditions were optimized according to the manufacturer's protocol.

## Puromycin incorporation

The aminonucleoside puromycin was added to the exponentially growing culture in YPD media at a final concentration of 10 μM for 15 min shaking at 30 °C, then 5 $OD_{600}$ of cells were collected, washed twice with $ddH_2O$ and prepared for immunoblot.

## Statistics of quantitative experimental data

All experiments were performed in at least three independent biological replicates. Visualization and statistical analysis was performed using the GraphPad Prism 9.0, unless otherwise stated. Two-tailed Student's $t$ test was used for the significance evaluation, unless otherwise stated.

## Reporting summary

Further information on research design is available in the Nature Research Reporting Summary linked to this article.

## Data availability

The proteome and transcriptome datasets are available in the PloidEx app as well as from public repositories. Transcriptome data have been deposited in the Gene Expression Omnibus database under accession code GSE162513. The mass spectrometry proteomics data have been deposited to the ProteomeXchange Consortium via the PRIDE partner repository with the accession code PXD022605. Source data are provided with this paper.

## Code availability

Source code and a compiled version of the supplementary plotting application PloiDEx are publicly available on GitHub: https://github.com/PMenges/PloiDEx.

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

## Acknowledgements
We thank the members of Storchova lab, as well as Simen Rød Sandve and Johannes M. Herrmann for discussions and support. We thank Jan-Eric Bökenkamp for his help with the statistical evaluation of the data. We thank Sara Vanessa Bernhard for her help with the human tetraploid cells and Robbie Loewith for providing the Sch9-as plasmid constructs. This research was funded by Rhineland-Palatine Research Initiative BioComp. G.Y. is a visiting scientist funded by Alexander von Humboldt Foundation (Georg Foster Stipend). P.S.A. is supported by research funding from the TU Nachwuchsring Kaiserslautern, Walter Benjamin-Programm and an Add-on fellowship from the Joachim Herz Foundation.

## Author contributions
G.Y. was involved in conceptualization, investigation, validation and funding acquisition and performed most of the experiments, P.M. and M.R. were involved in mass spectrometry, data analysis and curation, P.S.A. and D.A.N. contributed to the experiments, A.W. created the strains and contributed to mass spectrometry, D.S. and P.C. contributed the dynamic transcriptome analysis, N.K. and M.M. the mass spectrometry analysis; V.S. contributed to data analysis; Z.S. was involved in conceptualization, writing, supervision and funding acquisition. All authors read and commented on the manuscript.

## Funding

## Competing interests
The authors declare no competing interests.
