## [Peer Review File · Nature Communications]

Sublinear scaling of cellular proteome with ploidyReviewer #1 (Remarks to the Author):

In this paper, Yahya et al. set out to determine the proteome response to ploidy changes. They generated isogenic haploid, diploid, triploid and tetraploid (1N-4N) *Saccharomyces cerevisiae* strains and compared both the absolute scaling of the proteome and transcriptome, as well as differential protein and mRNA expression across ploidy changes using SILAC proteomics and comparative differential transcription analysis, respectively. They demonstrate that both mRNA and protein levels increase with ploidy, but in a non-linear fashion, an observation they name ploidy-dependent scaling. When analyzing genes that are differentially regulated by ploidy (ploidy-dependent regulation), they observed that whilst the transcriptome is overall "downregulated" (i.e. not linearly scaled) with ploidy, differential expression is almost exclusively visible at the proteome level. Enrichment analyses revealed that genes associated with ribosome biogenesis and translation are repressed with increasing ploidy, and the authors confirmed that not only certain ribosomal proteins, but also 25S and 5.8S rRNA are downregulated with increasing ploidy. They identified the transcriptional repressor Tup1 as a candidate mediator of ploidy-dependent scaling and showed that Tup1 abundance and stability increase with ploidy, negatively regulated through Tor1-Sch9 activity. Lastly, the authors tested their hypotheses in a human near-tetraploid cell line (HPT2) as an alternative model, confirming a similar effect of higher Tle1, the (human Tup1 ortholog, on the abundance of cellular rRNA levels and translation.

Overall, this is a very well-designed and highly interesting study showing solid evidence for a model proposing that Tup1 regulates ploidy-dependent scaling via downregulation of rDNA expression. It will additionally serve as a useful resource for other researchers. The manuscript would be improved by addressing the following issues:

Major points

- 1. The authors describe that increased ploidy in yeast cells leads to higher genome instability. How did the authors ensure that the observed effects were due to whole genome ploidy rather than to mixed aneuploid populations that often arise (at the different basal ploidy levels). Aneuploids generally induce stress that also leads to changes in the proteomes. How do these polyploid proteomes differ from those of aneuploid strains? This question applies also to the HPT2 human cell line, which the authors state to be hypo-tetraploid and heterogeneous. Are the effects there due to polyploidy or mixed aneuploidy?**
- 2. The authors show that Tup1 abundance increases with ploidy, and show strong evidence for a reduction of rRNA levels with higher ploidy. Since the scaling of the observed proteome is ploidy-dependent, presumably requiring more Tup1 to achieve higher levels of downregulation of ribosomes, is there direct evidence that rRNA levels scale with Tup1 expression? For example, do rRNA levels increase back to haploid levels if Tup1 is deleted/inhibited in cells with higher ploidy, or does increasing the expression of Tup1 lead to an equivalent reduction of rRNA levels? Were tRNA levels analyzed?**
- 3. How were immunoblots quantified? To what were they normalized? How was the linear range for these signals determined and do we know that the results were all within the linear range?**

Minor points

- 1. Fig 2c: The authors should use the same gradient for displaying the log₂FC determined via proteomics and the log₂FC determined in their immunoblotting validation experiments. Some of the immunoblotting results, namely for proteins Rfa1, Mrpl40, Nqm1, and Clb2 do not match the proteomics results. A discussion of this issue would be helpful.**
- 2. Antibodies for Rtg2, Tup1, Bre2, Rfa1, Nqm1, Rps23B and Rpl9B are not listed in Supplementary Table 2. Please add these.**
- 3. Fig 2c and SI Fig 6: It would be nice to see Volcano plots showing the protein and mRNA log₂ fold-changes vs the determined significance levels for the different ploidy**

comparisons to place the displayed selection of top/bottom 25 differentially regulated proteins into context.

4. Fig 2f, g: The authors state that they performed GSEA using YeastEnrichR and mention in their Materials and Methods that they “perform[ed] gene set enrichment analysis of statistically significantly different values of all ploidies to haploid.” Please specify what value was used to run the GSEA (.i.e., how was the degree of membership for the YeastEnrichR algorithm determined?)

5. Puromycin is an aminonucleoside and this term is preferable to “amino acid analog”.

6. Page 8 line 7. “We therefore examined all differentially regulated proteins for motives (sic) commonly found in repressors or for annotation indicating their involvement in rRNA” and identify Tup1 as a candidate rRNA repression gene. Please expand the description of the motif search performed: which motifs were used to search?, and was Tup1 the only candidate gene arising from the search?

7. Why was DCD-treated HCT116 chosen as a control for the near-tetraploid HPT2 cells? This is not evident and will benefit from an explanation.

8. Throughout the manuscript, magnitude identifiers such as “ μ ” for units did not print properly in the PDF received. Please check that all units are correct.

Reviewer #2 (Remarks to the Author):

Yahya et al. have submitted a manuscript on comprehensive investigation of cellular protein levels in yeast cells of different ploidy. Using a SILAC-based proteomics approach, they showed that protein (and mRNA) abundance scales allometrically with ploidy and that this scaling occurs via decreased ribosomal protein (and rRNA) abundance and reduced translation. The authors show that Tor1 activity is reduced with increasing ploidy, which leads to repression of the rRNA gene via a Tor1-Sch9-Tup1 signaling pathway. Furthermore, they show that mTORC1 and S6K activity are also reduced in human tetraploid cells, which points to a presence of a conserved pathway responsible for proteome remodeling in response to increased ploidy.

The manuscript is simple and well written, the topic is of fundamental importance. The initial SILAC is well designed: the authors have grown all cultures in the light SILAC medium and spiked the same heavy-labeled standard into all samples. This strategy is sometimes termed “Super-SILAC” and enables indirect quantitative comparison of >3 samples using a SILAC standard. Measurements of cell counts and volumes enabled accurate estimates of relative ploidy-dependent protein scaling, which was independently confirmed. The coverage of ca. 3100 protein groups was not particularly extensive, but more than enough to support all conclusions in the manuscript. Observed ploidy-dependent regulation of specific proteins and pathways (e.g. Tor1-Sch9-Tup1) was independently confirmed by other methods.

I support publication in Nat. Communications, but have two minor points:

1) Figure 4d: it would be better to determine the substrates of sch9 by MS rather than phos-tag gel (sch9-as is a nice tool for this purpose and results will be cleaner than phos-tag results shown)

2) Figure 5c: to claim that the observed phenomenon is conserved from yeast to human, it would be better to perform a proteome-wide study on cancer cells, rather than use western blot on a handful of markers

Reviewer #3 (Remarks to the Author):

The authors attempted to address the question of how yeast proteome changes with increasing ploidy and tried to suggest the downregulation of the mTOR signaling to be responsible for the reduced protein biosynthesis in polyploid cells. SILAC-based quantitative MS and comparative differential transcription analysis indicated both protein and concentration mRNA abundance did not scale linearly in a collection of MATa haploid to tetraploid strains. Some proteins and mRNAs were indicated to be differentially expressed based on their normalized Log2 fold change. Follow up GSEA analysis of proteins pointed to the possibility of downregulation translation, which could be linked to reduced rRNA abundance in cells with higher ploidy. The authors tried to zoom in the downregulation of the mTOR signaling, which could also be seen in a near-tetraploid cancer cell line.

Overall, the initial observation of ploidy-dependent protein scaling (PDS) is interesting. But the lack of rigorous and quantitative testing of the effects of ploidy on proteome and transcriptome at single cell level hampers understanding the potential impact of the study. Specific concerns and suggestions are listed below.

Major points:

1. As the authors indicated, tetraploid yeasts suffered from genome instability. The reiterative process of generating the many 4N strains used in the studies could lead to the accumulation of unwanted genetics changes that could contribute to 'PDS'. The authors shall provide genome resequencing data to check for gene/chromosomal mutations, in particular rDNA copy number changes.
2. The authors classified differentially regulated proteins with fold change cut off and some sort of 'smoothed filtering' (page 20). This does not seem appropriate; the details shall be listed in Material and Methods. Equally confusing is on the differential gene expression. The authors indicated in page 6 the use of 'log2FC relative to haploid levels' and 'normalized them by shifting the median of the distribution to 0.' But in the Material and Methods section in page 21, 'Differential gene expression analysis was performed using the R Bioconductor package Limma.' The authors should provide clarification on how the analysis was actually done.
3. How could ribosome biogenesis be Upregulated and Downregulated in Figure 2f?
4. The most intuitive expression of protein abundance is molecules per cell. For instance, there are approximately 42 million protein molecules in a haploid yeast (PMID: 29361465). Could the SILAC data here be converted into absolute quantification? Could the dilution in mRNA molecular account for the 'PDS' (Supplementary Fig. 3) besides rRNA? The use of absolute quantification of protein and mRNA abundances (PMID: 28365149) could help address this question.
5. Using single cell measurement, the authors calculated the median volumes of 48.0 fl for 1N, 82.9 fl for 2N, 146.6 fl for 3N and 181.7 fl for 4N. Along the same vein, single cell and single molecular counting based approach could be used to validate the differentially expressed genes or proteins. Validation is better done with counting GFP-tagged proteins molecules using fluorescence microscopy with in vivo standards, instead of the semi-quantitative Western Blot.
6. The authors suggested in page 10 'decreased surface-to-volume ratio may impair nutrient uptake, which in turn reduces mTOR activity and forces downregulation of protein synthesis.' But the result from cell size mutants Supplementary Figure 5 does not support the hypothesis. In the schematic model of the translation regulation in polyploid cells (Figure 5j), reduced import, membrane stress and mitochondrial stress was proposed to alter mTOR activity in 4N cells. The authors shall at least provide better discussion since no experimental data was shown in yeast or cancer cells.

Minor points:

1. The use of single protein loading control like Pgk1 in Figure 4c is not justified. Also the use the ACT1 as qPCR reference gene need to be justified.

- 2. Most Western Blot and protein gel images are of low resolution. It could be hard to tell if there was any change like in Supplementary Figure 8b.**
- 3. Cell size affects OD600. It's no appropriate to directly plot OD600 for cells of different ploidy without calibration in Supplementary Figure 1e.**
- 4. The title can be changed to 'Sublinear scaling of cellular proteome with ploidy in budding yeast.'**
- 5. Page 10, 'This first proteome analysis in cells of different ploidy reveals striking ploidy-dependent proteome changes.' Should be 'This first proteome analysis in yeast cells' as there are many others proteomic studies of polyploid plants, cancer cells etc.**
- 6. The abuse of Student T-test in multi group comparisons.**

REVIEWER COMMENTS

Reviewer #1 (Remarks to the Author):

In this paper, Yahya et al. set out to determine the proteome response to ploidy changes. They generated isogenic haploid, diploid, triploid and tetraploid (1N-4N) *Saccharomyces cerevisiae* strains and compared both the absolute scaling of the proteome and transcriptome, as well as differential protein and mRNA expression across ploidy changes using SILAC proteomics and comparative differential transcription analysis, respectively. They demonstrate that both mRNA and protein levels increase with ploidy, but in a non-linear fashion, an observation they name ploidy-dependent scaling. When analyzing genes that are differentially regulated by ploidy (ploidy-dependent regulation), they observed that whilst the transcriptome is overall “downregulated” (i.e. not linearly scaled) with ploidy, differential expression is almost exclusively visible at the proteome level. Enrichment analyses revealed that genes associated with ribosome biogenesis and translation are repressed with increasing ploidy, and the authors confirmed that not only certain ribosomal proteins, but also 25S and 5.8S rRNA are downregulated with increasing ploidy. They identified the transcriptional repressor Tup1 as a candidate mediator of ploidy-dependent scaling and showed that Tup1 abundance and stability increase with ploidy, negatively regulated through Tor1-Sch9 activity. Lastly, the authors tested their hypotheses in a human near-tetraploid cell line (HPT2) as an alternative model, confirming a similar effect of higher Tle1, the (human Tup1 ortholog, on the abundance of cellular rRNA levels and translation. Overall, this is a very well-designed and highly interesting study showing solid evidence for a model proposing that Tup1 regulates ploidy-dependent scaling via downregulation of rDNA expression. It will additionally serve as a useful resource for other researchers.

>> We thank the reviewer for the positive evaluation of our manuscript and for the constructive suggestions.

The manuscript would be improved by addressing the following issues:

Major points

1. The authors describe that increased ploidy in yeast cells leads to higher genome instability. How did the authors ensure that the observed effects were due to whole genome ploidy rather than to mixed aneuploid populations that often arise (at the different basal ploidy levels). Aneuploids generally induce stress that also leads to changes in the proteomes. How do these polyploid proteomes differ from those of aneuploid strains? This question applies also to the HPT2 human cell line, which the authors state to be hypo-tetraploid and heterogeneous. Are the effects there due to polyploidy or mixed aneuploidy?

>> This is a valid and important point and we apologize for not addressing this more carefully in the first submission. There are several reasons suggesting that aneuploidy does not play a role in this phenomenon. First, all our experiments were performed in newly made polyploids, to limit the number of cell divisions where mitotic errors leading to aneuploidy could occur. We collect the cells as soon as possible after validating the ploidy, freeze them and use always the same batch for experiments. During the experiments, we tested the ploidy by flow cytometry regularly. We now also performed additional analyses by flow cytometry of synchronized cells (Supplementary Fig. 1b) to further validate the ploidy. Second, we would like to point out that the allometric scaling of proteome was observed already in diploid cells (1.68x proteome of haploid cells, in comparison to expected 2x); there is no significant increase of chromosomal instability in diploids compared to

haploids. Additionally, to evaluate how the cellular response to increased ploidy compares to cellular response to aneuploidy, we compared the previously published transcriptome and proteome datasets from aneuploid yeast (obtained by Amon and Torres labs, PMID: 25073701). We found no overlap between polyploids and aneuploids in the pathway regulation or in protein levels of selected proteins (see the new Supplementary Fig. 6).

The possible effect of mixed aneuploidy is even more important in human cells where the tetraploid cells show marked chromosomal instability. To validate our previous results, we tested another near-tetraploid cell line derived from non-cancerous RPE1 cells. The near-tetraploid RPT1 contains 80 chromosomes (modal chromosome number) and immunoblotting of the ribosomal proteins revealed similar reduction as in HPT2. Additionally, we tested several HCT116 derived aneuploid cells with one extra chromosome, which we previously engineered by microcell-mediated chromosome transfer (e.g. PMID 31778112). While the protein abundance of ribosomal proteins is reduced in post-tetraploid cells, we did not observe comparable changes in trisomic cells (Supplementary Fig. 10 d,e). We conclude that aneuploidy induces phenotypic changes that are biologically and phenotypically distinct from changes in response to whole genome duplication. We agree, however, that the interpretation requires caution, because none of the analyzed post-tetraploid human cell lines maintains truly stable, tetraploid karyotype. We added this new data to the manuscript and amended the discussion.

2. The authors show that Tup1 abundance increases with ploidy, and show strong evidence for a reduction of rRNA levels with higher ploidy. Since the scaling of the observed proteome is ploidy-dependent, presumably requiring more Tup1 to achieve higher levels of downregulation of ribosomes, is there direct evidence that rRNA levels scale with Tup1 expression? For example, do rRNA levels increase back to haploid levels if Tup1 is deleted/inhibited in cells with higher ploidy, or does increasing the expression of Tup1 lead to an equivalent reduction of rRNA levels? Were tRNA levels analyzed?

>> As we showed in the first submission, deletion of Tup1 in haploid cells indeed leads to increased rRNA levels, corresponding increase in ribosome biogenesis and basal translation (Fig. 4f). We wanted to create a tetraploid strain lacking all four copies of the *TUP1* gene, but we have not succeeded so far. We managed only to delete two copies in 4N cells. This deletion resulted in increased cell size and increased translation (see figure R1 below). However, we struggled during these experiments with technical issues, because the deletion of *TUP1* leads to a strong flocculation, which was even increased in tetraploid cells and complicated cell counting (Tup1 is a known repressor of *FLO* genes, e.g. PMID: 25106892); moreover, the cellular morphology was altered and the cell size variability increased. We prefer not to add this data to the manuscript, because they are based only on heterozygous deletion and because the strong flocculation made the work with these cells difficult, leading to variable results. We are currently trying alternative approaches, which would be more technically robust and which would be part of our future study on the regulation of Tup1 in response to ploidy changes.

We have not analyzed the tRNA levels. We consider it beyond the scope of the current manuscript, but might evaluate this aspect in the future.

Figure R1 Effect of *TUP1* deletion on yeast tetraploid cells. (a) Yeast tetraploids lacking two of the four copies of *TUP1* show an increase in puromycin incorporation compared to wild type tetraploids. Two clones were tested, three independent experiments were performed, mean + SEM are shown. (b) Deletion of *TUP1* leads to increased cell size.

3. How were immunoblots quantified? To what were they normalized? How was the linear range for these signals determined and do we know that the results were all within the linear range?

>> Immunoblots were quantified relative to the corresponding Ponceau staining of the same membrane within the area of the membrane corresponding with the size of the analyzed protein. The abundance was normalized to 1N. In the cyclohexamide shut off experiments, the immunoblots were quantified relative to Pgk1, which was previously shown to be relatively stable over prolonged treatment with cyclohexamide; here the abundance was normalized to timepoint 0. The linear range was determined using the digital imaging software of the used imaging instrument Azure 500, AzureSpot Pro, which notifies when the acquired image has saturated bands. The software can determine the optimal image time that prevents signal saturation of the most intense bands. This feature makes sure that the data were within the linear range of the imaging system. We added this information to Material and Methods description.

Minor points

1. Fig 2c: The authors should use the same gradient for displaying the log₂FC determined via proteomics and the log₂FC determined in their immunoblotting validation experiments. Some of the immunoblotting results, namely for proteins Rfa1, Mrpl40, Nqm1, and Clb2 do not match the proteomics results. A discussion of this issue would be helpful.

>> We thank the reviewer for making us aware of this discrepancy. We corrected the gradient used for the plot. For the validation, we selected proteins of all categories – upregulated, downregulated, and unchanged with ploidy. While abundance changes could be very well validated for proteins showing strong ploidy-specific up- or downregulation (RTG2, TUP, MRPS51, ILV5 and RPS23B, RPL9B respectively), for other proteins the correlation between the proteomics and the immunoblotting is only modest. There might be several reasons why the results do not match perfectly. In high through-put proteomic studies, highly abundant proteins are much more accurately quantified than low abundant proteins. This is reflected by the increased standard deviations observed for the

biological replicates of low abundant proteins compared to highly expressed ones. This is an inherent problem, particularly if large abundance changes need to be quantified (e.g. for proteins that become detectable only in one of the compared states). However, robust statistical procedures which consider the variation between replicates, often compensate these limitations. Another factor limiting the accuracy of proteomic measurements is for example protein size, which determines the number of observable peptides per protein.

We noted that CLB2, NQM1 and MRPL40, are expressed at very low levels in yeast (CLB2: 56.2 kDa, 339 copies/cell; NQM1: 37.3 kDa, 4613 copies/cell; MRPL40: 33.7 kDa, 4845 copies/cell according to SGD). These proteins also show much lower correlation of proteomics and immunoblotting data. Consequently, some of the proteins pointed out by the reviewer (BRE2, CLB2) show a high variance and have many missing values in the proteomic data set (see figure R2). This likely accounts for the poor correlation observed between the proteomics and western blot analysis. We therefore decided to remove Bre2 and Clb2 from the validation. Additional discussion of the issue was added to the manuscript.

Figure R2 Re-evaluation of the proteomics data of proteins which were validated by immunoblotting. Median centered L/H intensity ratios were plotted against ploidy. Note, that Bre2 and Clb2 show many missing values and therefore were excluded from the final figure.

2. Antibodies for Rtg2, Tup1, Bre2, Rfa1, Nqm1, Rps23B and Rpl9B are not listed in Supplementary Table 2. Please add these.

>> We apologize for the lacking information. We have now added the information as requested. Rtg2, Tup1, Bre2, Rfa1, and Nqm1 are TAP-tag fusions from the yeast TAP-tagged library, and we used for all of them the PAP antibody. The information on Rps23B (Sc-100837, mouse, Santa Cruz) and Rpl9B (AP16409b-ev, Rabbit, ABGENT) antibodies was now also added.

3. Fig 2c and SI Fig 6: It would be nice to see Volcano plots showing the protein and mRNA log2 fold-changes vs the determined significance levels for the different ploidy comparisons to place the displayed selection of top/bottom 25 differentially regulated proteins into context.

>> We have now added a corresponding volcano plot to supplementary Figure S4e and highlighted the proteins shown in the corresponding heatmaps (now Fig. S4a-b and Fig. S7b (previously S6)).

We also calculated volcano plots for the proteins shown in Figure 2a-b and 2c (see Figure R3) but we do not want to include them in the manuscript. The reason is that the proteins with consistent ploidy specific expression changes across ploidy, are not necessarily those which show the highest intensity ratios between haploid and tetraploid states. To robustly identify PDR genes, we have now fitted a linear model to the data and report the corresponding coefficients for the slopes and p-values in the supplementary table Data1 (see also plots of selected proteins in Fig. R2 above).

Similarly, volcano plots highlighting proteins chosen for validation by Western blotting will not be essential for the main conclusions of the study and thus were omitted for clarity. These proteins were chosen due to the limited availability of antibodies and included few examples of up- and downregulated proteins of complexes that are further discussed in the manuscript, as well as a few proteins, which did not show strong PDR (e.g. RFA1, MDH1, MRPL40).

Figure R3 Volcano plots (a) The top 25 proteins showing a consistent trend of abundance changes across all ploidies and at least a 2FC change 4N/1N, as shown in Figure 2a,b. (b) Proteins validated by immunoblotting. For validation, we selected upregulated, downregulated and unchanged proteins.

4. Fig 2f, g: The authors state that they performed GSEA using YeastEnrichR and mention in their Materials and Methods that they “perform[ed] gene set enrichment analysis of statistically significantly different values of all ploidies to haploid.” Please specify what value was used to run the GSEA (i.e., how was the degree of membership for the YeastEnrichR algorithm determined?)

>> We apologize for not clearly explaining the method. Here, we used unranked up and

downregulated proteins identified in a multiple pooled t-tests (all ploidy against 1N). The details of the analysis were now added to Material and Methods. To independently determine ploidy specific pathway deregulation, we also performed GSEA using as input a protein list ranked according to the T-statistic derived from the linear model fitted to the ploidy state described above (Fig. R4, Supplementary Data1, Columns AG-AW). The GOBP category peptide metabolic process shows the highest normalized enrichment score among downregulated pathways with 93/338 members in the leading edge (Fig. R4b). Remarkably, more than two thirds of these proteins are subunits of the yeast cytosolic ribosome, confirming our main conclusions from our previous pathway enrichment analyses based on the comparisons of the proteomes from cells with higher ploidy to the one from haploid yeast cells. Consistent with our new analysis, the majority of the subunits of the cytoplasmic ribosome negatively correlated with ploidy according to our statistical model, whereas proteins of the mitochondrial ribosome do not (Fig. R4d). These new data are now also partly included in the Supplementary Fig. 9 b,c.

Figure R4 Pathway enrichment analysis based on the t-statistic of the ploidy-dependent linear model (a) GSEA was carried using the t-statistic derived from a linear model fitted to proteome LH ratios for the different ploidy states. Analysis was carried out using the online tool <http://www.webgestalt.org/> (b). Summary of the result for the GOBP category *peptide metabolic process*. (c) The WEBGESTALT score of each member of the GOBP category *peptide metabolic process* in the leading edge was plotted against the rank and ribosomal proteins were colored as indicated. (d) Volcano plot showing the coefficient2 plotted against the p-value derived from a model fitted to LH protein intensity ratios against ploidy. Subunits of the cytoplasmic and mitochondrial ribosome are highlighted in red and blue, respectively.

5. Puromycin is an aminonucleoside and this term is preferable to “amino acid analog”.

>>We thank the reviewer for this remark. We changed it in the manuscript

6. Page 8 line 7. “We therefore examined all differentially regulated proteins for motives (sic) commonly found in repressors or for annotation indicating their involvement in rRNA” and identify Tup1 as a candidate rRNA repression gene. Please expand the description of the motif search performed: which motifs were used to search?, and was Tup1 the only candidate gene arising from the search?

>> General transcriptional repressors are rich in repeats of Tryptophane-Aspartic Domain (WD or beta-transducin motives WD40) (doi: 10.1371/journal.ppat.1002235, DOI: 10.1038/369758a0, doi: 10.1371/journal.ppat.1002235). Among the proteins that have WD repeats and found to be differentially regulated relative to ploidy was Tup1. The mammalian homolog of Tup1, Tle1, is repressor of rRNA expression and contains the WD40 repeat motif as well (doi:10.1038/35057062). We deleted TUP1 in haploid budding yeast and found increased expression of rRNA. Subsequently, we continued with the analysis. The notion of the WD40 motive and the corresponding reference have now been added to the manuscript.

No other protein showed similar features as Tup1, but another interesting protein with a role in rRNA expression was identified during literature search: Irs4, protein involved in starvation response, cell wall organization, inositol lipid-mediated signaling, autophagy, and also reported to have an effect on rDNA silencing (shown in PMID: 10082585). Irs4 is downregulated with ploidy, although not very strongly (oppositely to Tup1, which is upregulated with ploidy), and deletion of Irs4 leads to reduced rRNA levels in haploid cells (Fig. R5). Only little is known about Irs4 (there are only 5 papers on Irs4 in budding yeast in PubMed search) and we do not know by what mechanism Irs4 affects the rRNA abundance. We continue working on this aspect, but decided against adding it to the manuscript. Nevertheless, the notion that Irs4 abundance shows opposite trend than Tup1 in polyploid cells, and has opposite effect on rRNA levels and translation provides additional support to our hypothesis.

Figure R5 Effect of the deletion of the *IRS4* gene on rDNA expression (a) and on puromycin incorporation as a proxy of translation efficiency (b).

7. Why was DCD-treated HCT116 chosen as a control for the near-tetraploid HPT2 cells? This is not evident and will benefit from an explanation.

>> We apologize for the missing explanation. Dihydrocytocholasin D is an inhibitor of actin polymerization and the treatment leads to cytokinesis failure and formation of tetraploid cells. In this case we intended to create *de novo* tetraploids. *De novo* arising tetraploids have a 4N DNA content,

but many of the cells arrest soon after the genome doubling. We now added the explanation to the text.

8. Throughout the manuscript, magnitude identifiers such as “ μ ” for units did not print properly in the PDF received. Please check that all units are correct.

>>We apologize for this error, which we could not see in our files. We hope that in the new manuscript version the conversion to pdf occurred without any similar errors.

Reviewer #2 (Remarks to the Author):

Yahya et al. have submitted a manuscript on comprehensive investigation of cellular protein levels in yeast cells of different ploidy. Using a SILAC-based proteomics approach, they showed that protein (and mRNA) abundance scales allometrically with ploidy and that this scaling occurs via decreased ribosomal protein (and rRNA) abundance and reduced translation. The authors show that Tor1 activity is reduced with increasing ploidy, which leads to repression of the rRNA gene via a Tor1-Sch9-Tup1 signaling pathway. Furthermore, they show that mTORC1 and S6K activity are also reduced in human tetraploid cells, which points to a presence of a conserved pathway responsible for proteome remodeling in response to increased ploidy.

The manuscript is simple and well written, the topic is of fundamental importance. The initial SILAC is well designed: the authors have grown all cultures in the light SILAC medium and spiked the same heavy-labeled standard into all samples. This strategy is sometimes termed “Super-SILAC” and enables indirect quantitative comparison of >3 samples using a SILAC standard. Measurements of cell counts and volumes enabled accurate estimates of relative ploidy-dependent protein scaling, which was independently confirmed. The coverage of ca. 3100 protein groups was not particularly extensive, but more than enough to support all conclusions in the manuscript. Observed ploidy-dependent regulation of specific proteins and pathways (e.g. Tor1-Sch9-Tup1) was independently confirmed by other methods.

I support publication in Nat. Communications, but have two minor points:

1) Figure 4d: it would be better to determine the substrates of sch9 by MS rather than phos-tag gel (sch9-as is a nice tool for this purpose and results will be cleaner than phos-tag results shown)

>> We thank the reviewer for the positive evaluation and the constructive criticism. While we agree that sch9-as is a great tool, MS of the global proteome would not provide a direct evidence of phosphorylation of Tup1 by Sch9, because other intermediate kinases could be involved in the phosphorylation. Instead, we planned to test directly whether Sch9 phosphorylates Tup1. We initiated experiments to purify Sch9 and Tup1 and perform *in vitro* phosphorylation experiments. While purification of Sch9 and Sch9-as was successful, we were not able to purify Tup1 so far. We however hope to overcome these issues in the future. Currently, we are convinced that these experiments are beyond the scope of the manuscript, but plan to perform these experiments to characterize better the Sch9-Tup1 pathway.

2) Figure 5c: to claim that the observed phenomenon is conserved from yeast to human, it would be better to perform a proteome-wide study on cancer cells, rather than use western blot on a handful of markers

>> We agree with the reviewer that the statement is too general and not justified by the results. A proteome wide study of human tetraploid cells is planned, but this is a more complex endeavor, which is beyond the scope of this manuscript (but planned for the future). Therefore, we rephrased the statement and pointed out the limitations of our study.

Reviewer #3 (Remarks to the Author):

The authors attempted to address the question of how yeast proteome changes with increasing ploidy and tried to suggest the downregulation of the mTOR signaling to be responsible for the reduced protein biosynthesis in polyploid cells. SILAC-based quantitative MS and comparative differential transcription analysis indicated both protein and concentration mRNA abundance did not scale linearly in a collection of MATa haploid to tetraploid strains. Some proteins and mRNAs were indicated to be differentially expressed based on their normalized Log2 fold change. Follow up GSEA analysis of proteins pointed to the possibility of downregulation translation, which could be linked to reduced rRNA abundance in cells with higher ploidy. The authors tried to zoom in the downregulation of the mTOR signaling, which could also be seen in a near-tetraploid cancer cell line.

Overall, the initial observation of ploidy-dependent protein scaling (PDS) is interesting. But the lack of rigorous and quantitative testing of the effects of ploidy on proteome and transcriptome at single cell level hampers understanding the potential impact of the study. Specific concerns and suggestions are listed below.

Major points:

1. As the authors indicated, tetraploid yeasts suffered from genome instability. The reiterative process of generating the many 4N strains used in the studies could lead to the accumulation of unwanted genetics changes that could contribute to 'PDS'. The authors shall provide genome resequencing data to check for gene/chromosomal mutations, in particular rDNA copy number changes.

>> This is an important and valid point and we apologize for not addressing this more carefully on the first submission. In particular, we consider the rDNA copy number changes very important and thank the reviewer for this suggestion. To address this point, we performed qPCR of the genomic DNA using a method from PMID 26195783. These experiments showed that there are no significant differences between the cells of different ploidy (Supplementary Fig. 9d).

While the idea of resequencing of the strains is interesting, we doubt that it would be helpful for our experiments. All our experiments were performed in newly made polyploids. We collect the cells as soon as possible after validating the ploidy, freeze them and use always the same batch for experiments. In fact, the construction of strains of different ploidy does not include more culturing than any other strain construction (e.g. deletion of a gene, or tagging of a gene). It is not standard in the field to resequence the entire genome after every strain construction. Second, we have constructed the cells of different ploidy several times independently (either as a control, or because each time we tag a protein for WB or IF in haploid cells, we have to construct all higher ploidy strains again), each time with the same results – if this would be a result of random genetics changes, it would be difficult to imagine that the outcome would always be the same. Third, the proteome scaling is observed already in diploid cells (1.68x proteome of haploid cells, in comparison to expected 2x); there is no significant increase of genomic instability in diploids compared to haploids,

nor any long periods of culturing. Thus, we do not expect that aneuploidy or point mutations contribute to the observed phenomenon. To support this conclusion (also requested by Rev. 1) we performed additional analyses, which we added to the manuscript. We added additional cell cytometry of synchronized cells (Supplementary Fig. 1b) to further validate the ploidy. Also, comparison of the previously published transcriptome and proteome datasets from aneuploid yeast (obtained by Amon and Torres labs, PMID: 25073701) showed no overlap between polyploids and aneuploids in the pathway regulation or in protein levels of selected proteins (see the new Supplementary Fig. 6). Similarly, we tested additional near-tetraploid cell line RPT1, which was derived from RPE1 cells (non-cancerous human cell line), with similar results as observed in HCT116-derived tetraploids.

2. The authors classified differentially regulated proteins with fold change cut off and some sort of 'smoothed filtering' (page 20). This does not seem appropriate; the details shall be listed in Material and Methods. Equally confusing is on the differential gene expression. The authors indicated in page 6 the use of 'log2FC relative to haploid levels' and 'normalized them by shifting the median of the distribution to 0.' But in the Material and Methods section in page 21, 'Differential gene expression analysis was performed using the R Bioconductor package Limma.' The authors should provide clarification on how the analysis was actually done.

>>We apologize for the insufficient description of the methods. We have now expanded the description of the methods and clarified the used approaches.

The LIMMA package was used only for the transcriptome analysis, as is indicated in the Methods. We now amended the Material and Methods to avoid any confusion.

3. How could ribosome biogenesis be Upregulated and Downregulated in Figure 2f?

>> The problem here is that the GOBP term "*ribosome biogenesis*" includes many structural ribosomal proteins as well as proteins involved in the biogenesis of ribosome alone. The Upregulated and Downregulated proteins are non-overlapping and the downregulated are enriched for structural ribosomal proteins.

Seeking an alternative strategy to analyze the data (also suggest by reviewer #1), we performed Gene Set Enrichment Analysis using as input a protein list ranked either by combined q-valued or according to the T-statistic derived from the linear model fitted to the ploidy state described above (Fig. R4, Supplementary table Columns AG - AW). By this method, the ribosome biogenesis pathway and translation were downregulated with increasing ploidy, no upregulation was identified. The GOBP category peptide metabolic process shows the highest normalized enrichment score among downregulated pathways with 93/338 members in the leading edge (Fig. R4b). Remarkably, more than two thirds of these proteins are subunits of the yeast cytosolic ribosome, confirming our main conclusions from our previous pathway enrichment analyses based on the comparisons of the proteomes from cells with higher ploidies to the one from haploid yeast cells. Consistent with our new analysis, the majority of the subunits of the cytoplasmic ribosome negatively correlated with ploidy according to our statistical model, whereas proteins of the mitochondrial ribosome do not (Fig. R4d, see above). Taken together, we have shown by three different statistical methods that ribosome biogenesis and translation is downregulated with increasing ploidy; this downregulation occurs only on proteome level. This new data confirm the previous results and are now shown in Fig. 2f and Supplementary Fig. 9 b, c.

4. The most intuitive expression of protein abundance is molecules per cell. For instance, there are approximately 42 million protein molecules in a haploid yeast (PMID: 29361465). Could the SILAC data here be converted into absolute quantification? Could the dilution in mRNA molecular account for the 'PDS' (Supplementary Fig. 3) besides rRNA? The use of absolute quantification of protein and mRNA abundances (PMID: 28365149) could help address this question.

>> Unfortunately, our current data set cannot be converted into absolute quantification, but we agree that in future this should be one aim of our experiments. The dilution in mRNA molecular most likely contributes to the PDS, as we pointed out previously and as we now further emphasized in the Discussion of our manuscript. The mRNA synthesis and degradation rate scales with ploidy, but strikingly, the scaling differs. In future, we want to evaluate the relative contribution of the transcription and translation regulation to the observed scaling.

5. Using single cell measurement, the authors calculated the median volumes of 48.0 fl for 1N, 82.9 fl for 2N, 146.6 fl for 3N and 181.7 fl for 4N. Along the same vein, single cell and single molecular counting based approach could be used to validate the differentially expressed genes or proteins. Validation is better done with counting GFP-tagged proteins molecules using fluorescence microscopy with in vivo standards, instead of the semi-quantitative Western Blot.

>> While this is an interesting idea, there are several caveats to these experiments. Indeed, imaging (or flow cytometry) would provide a single cell resolution, but it is questionable whether the (semi)-quantitative fluorescence imaging is more reliable than WB or mass spectrometry. In fact, it is not readily used for protein quantification. Many aspects affect the reliability of the quantitative imaging - protein folding as well as chromophore maturation efficiency, protein localization, photobleaching, autofluorescence, and more. Therefore, we prefer to use immunoblotting for validation.

6. The authors suggested in page 10 'decreased surface-to-volume ratio may impair nutrient uptake, which in turn reduces mTOR activity and forces downregulation of protein synthesis.' But the result from cell size mutants Supplementary Figure 5 does not support the hypothesis. In the schematic model of the translation regulation in polyploid cells (Figure 5j), reduced import, membrane stress and mitochondrial stress was proposed to alter mTOR activity in 4N cells. The authors shall at least provide better discussion since no experimental data was shown in yeast or cancer cells.

>> We thank the reviewer for this comment. While our data do not support the hypothesis that surface-to-volume ratio affects the phenotype, we also cannot fully exclude this option based on our evidence. A global scale proteomic approach in cells of different sizes would need to be performed to exclude the role of cell size fully. As suggested, we have now changed the Discussion on this topic and clarified why we still consider this possibility in our model.

Minor points:

1. The use of single protein loading control like Pgk1 in Figure 4c is not justified. Also the use the ACT1 as qPCR reference gene need to be justified.

>>Actually, Pgk1 is used frequently as a loading control for cyclohexamide shut off experiments, because it is a protein whose abundance does not vary over prolonged period of time of cyclohexamide treatment (PMID: 27167179). ACT1 mRNA is frequently used as qPCR reference. We validated that the ACT1 mRNA abundance does not change significantly with ploidy (only very few

mRNAs change with ploidy). Additionally, each sample was spiked with TATAA Universal RNA Spike II control (TATAA Biocenter AB, Sweden). The results were comparable

2. Most Western Blot and protein gel images are of low resolution. It could be hard to tell if there was any change like in Supplementary Figure 8b.

>> We apologize for the low quality which was largely caused by converting the figures into small size pdfs. We hope that the standard figures will provide sufficient resolution.

3. Cell size affects OD600. It's no appropriate to directly plot OD600 for cells of different ploidy without calibration in Supplementary Figure 1e.

>> Indeed, OD600 is influenced by the size as well as by the shape of the cells. We calibrated the OD to cell number for individual ploidies, and added it to the Supplementary Fig. 1f. Additionally, we added cautionary note to the figure legend.

4. The title can be changed to 'Sublinear scaling of cellular proteome with ploidy in budding yeast.'

>> We thank the reviewer for this suggestion. Indeed, this title captures more precisely the main finding of our research and therefore we decided to change the title.

5. Page 10, 'This first proteome analysis in cells of different ploidy reveals striking ploidy-dependent proteome changes.' Should be 'This first proteome analysis in yeast cells' as there are many others proteomic studies of polyploid plants, cancer cells etc.

>> Actually, we are not aware of any quantitative, proteome wide assessment of global scaling of proteins to ploidy in other species. However, we agree that the sentence should be changed and restated now as follows: "The presented proteome analysis in cells of different ploidy reveals striking ploidy-dependent proteome changes."

6. The abuse of Student T-test in multi group comparisons.

>> We now expanded our statistical analysis and calculated the linear models fitted to the ploidy state and the respective T-statistic (Supplementary Data 1, Columns AG-AW). This approach did not lead to different results. For example, the most downregulated pathway identified by using the T-statistic is strongly enriched for ribosomal proteins (Fig. R4, Supplementary Fig. 9b,c).

Reviewer #3 (Remarks to the Author):

The authors addressed my concerns and significantly improved the manuscript.

The editor asked me to provide feedback on the authors' responses to Reviewer #1.

I think the authors' response to the main point 2 of Reviewer #1 is unsatisfactory.

Reviewer #1 asked a critical question regarding the proposed molecular mechanism of ploidy-dependent sublinear scaling. Specifically, the authors did not address "Does increasing the expression of Tup1 lead to an equivalent reduction on rRNA levels?" While these are additional results on Tup1 deletion, the authors did not show or explain any data on the effect of increasing Tup1 dosage on rRNA levels. I would appreciate if the authors could add this important piece of evidence.

Typos

Page 3: "two sets of chromosomes"

Page 4: "The minimal changes observed in transcriptome raise"

Page 6: "we observed that several proteins were differentially regulated in cells of different ploidy"

Page 8: "deletion of TUP1 in 1N strains led to"

Page 9: "loss of TUP1 led to constitutively high rRNA levels"

Reviewer #3 (Remarks to the Author):

The authors addressed my concerns and significantly improved the manuscript.

The editor asked me to provide feedback on the authors' responses to Reviewer #1.

I think the authors' response to the main point 2 of Reviewer #1 is unsatisfactory.

Reviewer #1 asked a critical question regarding the proposed molecular mechanism of ploidy-dependent sublinear scaling. Specifically, the authors did not address "Does increasing the expression of Tup1 lead to an equivalent reduction on rRNA levels?" While these are additional results on Tup1 deletion, the authors did not show or explain any data on the effect of increasing Tup1 dosage on rRNA levels. I would appreciate if the authors could add this important piece of evidence.

>>Unfortunately, we are currently not able to perform these experiments. Upon discussion with the editor we pointed out in the Discussion that additional experiments would be required:

"How exactly Tup1 affects the rRNA abundance in budding yeast, and whether the effect is direct or indirect, needs to be addressed in the future. To this end, the impacts of Tup1 overexpression on rRNA synthesis and processing, as well as the binding of Poll at the ribosomal DNA promoter in cells of different ploidy, should be analyzed."

Typos

Page 3: "two sets of chromosomes"

Page 4: "The minimal changes observed in transcriptome raise"

Page 6: "we observed that several proteins were differentially regulated in cells of different ploidy"

Page 8: "deletion of TUP1 in 1N strains led to"

Page 9: "loss of TUP1 led to constitutively high rRNA levels"

>>We thank the reviewer for the careful reading. We edited the manuscript accordingly.